# Drivers and modelling of blue carbon stock variability in sediments of southeast Australia

Carolyn J. Ewers Lewis[1], Mary A. Young[2], Daniel Ierodiaconou[2], Jeffrey A. Baldock[3], Bruce Hawke[3], Jonathan Sanderman[4], Paul E. Carnell[1], Peter I. Macreadie[1]

[1]School of Life and Environmental Sciences, Centre for Integrative Ecology, Deakin University, 221 Burwood Highway, Burwood, Victoria 3125, Australia

[2]School of Life and Environmental Sciences, Centre for Integrative Ecology, Deakin University, Princes Highway, Warrnambool, Victoria 3280, Australia

[3]Commonwealth Scientific and Industrial Organisation, Agriculture and Food, PMB 2, Glen Osmond, South Australia 5064, Australia

[4]Woods Hole Research Center, 149 Woods Hole Road, Falmouth MA 02540 USA.

*Correspondence to:* Carolyn J. Ewers Lewis (ce8dp@virginia.edu)

**Abstract.** Tidal marshes, mangrove forests, and seagrass meadows are important global carbon (C) sinks, commonly referred to as coastal 'blue carbon'. However, these ecosystems are rapidly declining with little understanding of what drives the magnitude and variability of C associated with them, making strategic and effective management of blue C stocks challenging. In this study, our aims were threefold: 1) identify ecological, geomorphological, and anthropogenic variables associated with 30-cm deep sediment C stock variability in blue C ecosystems in southeast Australia; 2) create a predictive model of 30-cm deep sediment blue C stocks in southeast Australia; and, 3) map regional 30-cm deep sediment blue C stock magnitude and variability. We had the unique opportunity of using a high-spatial-density C stock dataset of sediments to 30 cm deep from 96 blue C ecosystems across the state of Victoria, Australia, integrated with spatially explicit environmental data to reach these aims. We used an information theoretic approach to create, average, validate, and select the best averaged general linear mixed effects model for predicting C stocks across the state. Ecological drivers (i.e. ecosystem type or dominant species/ecological vegetation class) best explained variability in C stocks, relative to geomorphological and anthropogenic drivers. Of the geomorphological variables, distance to coast, distance to freshwater, and slope best explained C stock variability. Anthropogenic variables were of least importance. Our model explained 46% of the variability in 30-cm deep sediment C stocks and we estimated over 2.31 million Mg C stored in the top 30 cm of sediments in coastal blue C ecosystems in Victoria, 88% of which was contained within four major coastal areas due to the extent of blue C ecosystems (~87% of total blue C ecosystem area). Regionally, these data can inform conservation management, paired with assessment of other ecosystem services, by enabling identification of hotspots for protection and key locations for restoration efforts. We recommend these methods be tested for applicability to other regions of the globe for identifying drivers of sediment C stock variability and producing predictive C stock models at scales relevant for resource management.

**1 Introduction**

Vegetated coastal wetlands – particularly tidal marshes, mangrove forests and seagrass meadows – serve as valuable organic carbon (C) sinks, earning them the term 'blue carbon' (Nellemann et al., 2009). Still, an increasing proportion of these ecosystems are being degraded and converted, and with pressures associated with human population growth the competition for land use in coastal zones continues to increase. With the current momentum for including blue C ecosystems in global greenhouse gas inventories, there is a need to quantify the

magnitude of C stocks and fluxes, especially in the sediments where the majority of the long-term C pool persists (Mcleod et al., 2011). However, global and regional assessments of blue C reveal large variability in sediment C stocks, both on small and large scales (Ewers Lewis et al., 2018; Liu et al., 2017; Macreadie et al., 2017a; Ricart et al., 2015; Sanderman et al., 2018). Identification of environmental variables driving differences in sediment C stocks in blue C ecosystems has become a key objective in blue C science and a necessary next step for quantifying

C storage as an ecosystem service. Knowledge of such drivers is also important for coastal blue C management, including identification of hotspots to prioritize for conservation, as well as maximization of C gains through strategic restoration efforts.

Drivers of sediment C stock variability are innately difficult to identify in that the stocks represent the net result of many complex processes acting simultaneously, simplified as: 1) production of autochthonous C; 2)

trapping and burial of autochthonous and allochthonous C, and; 3) remineralization and preservation of buried and surface C. Spatial variability in sediment blue C stocks resulting from these processes exists in hierarchical levels across global, regional, local, and ecosystem patch level scales (Ewers Lewis et al., 2018; Sanderman et al., 2018) and may be influenced by climatic, ecological, geomorphological, and anthropogenic factors (Osland et al., 2018; Rovai et al., 2018; Twilley et al., 2018).

At the global scale, climatic parameters appear to drive broad-scale variability in C stocks through effects on C sequestration (Chmura et al., 2003). Mangroves in the tropics have higher C stocks compared to subtropical and temperate mangroves, with rainfall being the single greatest predictor; when modelled, a combination of temperature, tidal range, latitude, and annual rainfall explained 86% of the variability in global mangrove forest C (Sanders et al., 2016). Sanderman et al. (2018) found large-scale factors driving soil formation (e.g. parent

material, vegetation, climate, relief) were four times more important than local drivers for predicting mangrove sediment C stock density; but still, localized covariates were necessary for modelling the variability of sediment C stocks at finer spatial scales.

Differences in sediment stocks have also been observed across blue C ecosystem types, with meter-deep C stocks being highest in tidal marshes (389.6 Mg C ha$^{-1}$), followed by mangroves (319.6 Mg C ha$^{-1}$), then seagrass

(69.9 Mg C ha$^{-1}$; Siikamäki et al. 2013). In southeast Australia this trend was observed on a regional scale, where an assessment of 96 blue C ecosystems revealed sediment C stocks to 30 cm deep were highest in tidal marshes (87.1 ± 4.9 Mg C ha$^{-1}$) and mangroves (65.6 ± 4.2 Mg C ha$^{-1}$), followed by seagrasses (24.3 ± 1.8 Mg C ha$^{-1}$; Ewers Lewis et al. 2018).

Considerable variability in sediment C stocks has also been observed across species of vegetation. Lavery

et al. (2013) compared 17 Australian seagrass habitats encompassing 10 species and found an 18-fold difference in sediment C stocks across them. Similarly, saltmarsh species differ not only in magnitude of C stocks, but also in their capacity to retain allochthonous C (Sousa et al. 2010a). Species richness within an ecosystem type may

also play a role in sediment C stock variability. In a global assessment, mangrove stands with five genera had 70-90% higher sediment C stocks per unit area compared to other richness levels (1-7 species stands; Atwood et al., 2017).

Beyond vegetation type, geomorphological factors appear to be most important when considering fine spatial scale sediment C stock variability (Sanderman et al., 2018). Elevation is likely an important driver of C stock variability in blue C ecosystems. Generally, the majority of the variability in C sequestration rates is linked to differences in sediment supply and inundation (Chmura et al. 2003). In lower elevations, faster sediment deposition may aid in C sequestration by trapping organic matter from macrophytes and microbes growing on soil surfaces (Connor et al. 2001). In higher elevations tidal flooding is less frequent, providing less opportunity for particles and C to settle out of the water column, resulting in a lower contribution of allochthonous C from marine or other sources compared to lower, more frequently inundated marshes (Chen et al., 2015; Chmura et al., 2003; Chmura and Hung, 2004).

The relative importance of elevation on sediment C stocks may vary depending on the contributions of autochthonous and allochthonous C. In ecosystems where the majority of the sediment C pool is autochthonous, elevation may be less important. Large variations in the origin of organic C can occur in mangroves, often with high C stocks being associated with autochthonous C and lower C stocks being associated with imported allochthonous C from marine and estuarine sources; similar variability in C origin has been observed in temperate tidal marshes (Bouillon et al. 2003). Higher C accumulation rates have been observed for upper tidal marsh assemblages that included rush (*Juncus*) compared to succulent (*Sarcocornia*) and grass (*Sporobolus*) tidal marsh assemblages located lower in the tidal frame (Kelleway et al., 2017). Rushes had high autochthonous C inputs, while sedimentation in succulents and grasses were mainly mineral.

Evidence is mounting that blue C ecosystems higher up in catchments (i.e. primarily fluvially influenced) maintain larger sediment C stocks than ecosystems further down in catchments (i.e. primarily marine influenced). For example, in southeast Australia, tidal marshes in brackish fluvial environments had sediment C stocks two times higher than those in marine tidal settings (Kelleway et al., 2016; Macreadie et al., 2017a). The deeper, stable C stores of tidal marshes are also higher in fluvial vs. marine-influenced settings, aiding long-term preservation of C (Van De Broek et al., 2016; Saintilan et al., 2013). The influence of fluvial inputs on sediment C stocks appears to be linked to three possible mechanisms: 1) fluvial environments are usually associated with smaller grain size sediments (silts and muds), which can enhance C preservation by reducing sediment aeration compared to sandy sediments (Kelleway et al., 2016; Saintilan et al., 2013); 2) higher freshwater input may lead to higher plant biomass and therefore autochthonous C inputs (Kelleway et al., 2016); and, 3) there is a greater contribution of terrestrial sediments via suspended particulate organic C and suspended sediment concentration higher up in the catchment compared to near the coast (Van De Broek et al., 2016).

Along with position in an estuary or catchment, proximity to freshwater inputs may drive differences in sediment C stocks among and within ecosystem patches. Tidal marsh accretion rates, which have been positively correlated (87%) with organic matter inventory, tend to decrease with distance from freshwater channels (Chmura and Hung, 2004), suggesting sediment C stocks may be higher closer to channels. Distance to freshwater is positively correlated with surface elevation, suggesting areas further from channels are inundated less frequently so have less sedimentation and slower accretion rates (Chmura and Hung, 2004).

It is important to note that high sedimentation rates do not necessarily result in high C sequestration rates or stocks if inorganic sediments make up a substantial portion of new sediment composition. Finer particles have higher surface area to volume ratios and tend to bind more organic molecules than coarse particles (Mayer, 1994). In seagrasses, high mud content is correlated with high sediment organic C content, except when large autochthonous inputs (e.g. seagrass detritus from large species such as those of *Posidonia* and *Amphibolis* genera) disrupt this correlation (Serrano et al., 2016a).

Anthropogenic activities may also influence the C sink capacity of blue C ecosystems, even when the sediments are not directly disturbed (Lovelock et al., 2017). Land use, particularly greater area of farmland and urbanization, has been associated with worsening of seagrass condition, including abundance and species richness (Quiros et al., 2017), which may result in impacts to sediment C stocks. Nutrient additions resulting from agriculture and urbanization may increase primary productivity in nutrient limited areas (Armitage and Fourqurean, 2016). However, reduced nutrient inputs to coastal ecosystems could benefit C sequestration, as nutrient additions can result in net C loss through plant mortality, erosion, efflux, and remineralization via enhanced microbial activity (Macreadie et al., 2017b). Further, excess N has been linked to enhanced decomposition and an overall increase in tidal marsh ecosystem respiration due to shifts in microbial communities (Kearns et al., 2018).

Land use and human population may also impact blue C sediment stocks through erosion of terrestrial soils. Human activities causing erosion on land can result in increased sediment loads to coastal areas, including fine particles with a high affinity for C (Mazarrasa et al., 2017; Serrano et al., 2016b). An average of 60% of global soil erosion has been tied to human activities, particularly population density, agriculture, and deforestation (Yang et al., 2003). Export of fine sediments to coastal ecosystems from eroded terrestrial soils may encourage trapping and preservation of C within the sediments of blue C ecosystems.

Assessments of the drivers of blue C stock variability are often completed at global scales (Atwood et al., 2017; Rovai et al., 2018). Given the variability of sediment C stocks at finer spatial scales, and that coastal resources are managed on finer scales, we wanted to investigate drivers influencing regional blue C sediment stock variability. Here, we had the opportunity to exclude comparisons between temperate and tropical climates or effects of latitude by working on a stretch of coastline that spans approximately 1500 km west to east. We tested the relationship between ecological, geomorphological, and anthropogenic variables and sediment blue C stocks in the mineral-dominated sediments of southeast Australia. By identifying drivers of small-scale variability in sediment C stocks, across and within ecosystem patches, we created a predictive model for estimating C stocks on a scale relevant to coastal resource management. Our specific objectives were to: 1) identify ecological, geomorphological, and anthropogenic factors driving variability in 30-cm deep sediment blue C stocks within and across ecosystem patches in southeast Australia, 2) produce a spatially explicit model of current 30-cm deep sediment blue C stocks based on the relative importance of environmental drivers in southeast Australia, and, 3) map regional 30-cm deep sediment blue C stock magnitude and variability.

## 2 Materials and Methods

### 2.1 Sediment C stock dataset

Sediment C stocks to 30 cm deep were estimated for 287 sediment cores from 96 blue C ecosystems
across Victoria in southeast Australia (Ewers Lewis, 2020; Ewers Lewis et al., 2018; Figure 1). Full details of
sample collection, laboratory analyses, and calculations of C stocks can be found in Ewers Lewis et al. (2018).
Briefly, three replicate sediment cores (5-cm inner diameter) were taken in each ecosystem (n=125 in tidal marsh,
n=60 in mangroves, and n=102 in seagrasses). Once back in the laboratory, samples were taken from three depths
(0-2, 14-16, 28-30 cm) within each core. Samples were dried at 60℃ until a consistent weight was achieved, then
ground. Dry bulk density (DBD) was calculated as the dry weight divided by the original volume for all samples.

Based on the protocols by Baldock et al. (2013), a combination of diffuse reflectance Fourier transform
mid-infrared (MIR) spectroscopy and elemental analysis via oxidative combustion using a LECO Trumac CN
analyzer was used to determine organic C contents of all samples.Previous studies have demonstrated the accuracy
of using MIR to estimate organic C stocks of sediments (Baldock et al., 2013; Van De Broek and Govers, 2019;
Ewers Lewis et al., 2018). MIR spectra were acquired for all samples, then a subset of 200 representative samples
was selected based on a principle components analysis (PCA) of the MIR results utilizing the Kennard-Stone
algorithm. Gravimetric contents of organic carbon were measured directly in the laboratory for the 200-sample
subset (Baldock et al. 2013). A partial least squares regression (PSLR) was created using a Random Cross
Validation Approach (Unscrambler 10.3, CAMO Software AS, Oslo, Norway) and used to build algorithms to
predict square root transformed total carbon, total organic carbon, total nitrogen, and inorganic carbon for the
entire dataset. The PSLR model was evaluated based on  parameters from the chemometric analysis of soil
properties (Bellon-Maurel et al., 2010; Bellon-Maurel and McBratney, 2011), and the relationship between
measured and predicted values was assessed based on slope, offset, correlation coefficient (r), R-squared, the root
mean square error (RMSE), bias, and the standard error (SE) of calibration (SEC) and validation (SEP; see Ewers
Lewis et al., 2018 for full details). R-squared values for all square root transformed variables were ≥0.94.

Sediment C stocks were calculated based on Howard et al., 2014. Organic C density (mg C cm$^{-3}$) was
calculated by multiplying organic C content (mg C g$^{-1}$) by DBD (g cm$^{-3}$). Linear splines were applied to each core
to estimate C density for each 2 cm increment within the 30 cm core, then C densities for each interval (measured
and extrapolated) were summed and converted to Mg C ha$^{-1}$ to estimate total stock down to 30 cm deep for each
core location.

Though it is common in the literature to sample to 1 m deep in blue C sediments, the sampling protocol
used for collecting these data (Ewers Lewis et al., 2018) was designed to maximize spatial coverage of sediment
C samples rather than sample entire sediment profiles (which may extend well beyond 1 meter deep). Greater
spatial coverage allowed us to test the relationships between a variety of potential drivers and 30-cm deep sediment
C stocks on both fine and broad scales.

**2.2 Generation of predictor variables**

Our general approach to identifying potential drivers of 30-cm deep sediment C stock variability was to
develop a predictive model based on spatially explicit environmental factors associated with our high spatial
density of sediment C sampling. For clarity, we have grouped predictor variables into three categories – ecological,
anthropogenic, and geomorphological – though the processes impacting C storage for each may span all three
categories (Table 1; Table S1).

Values of predictor variables for each core were determined from spatial data either as the collective value representing activities within the catchment or based on the exact location of sample collection, dependent on the variable. Geographical boundaries for catchments in Victoria were derived using high resolution elevation data and flow accumulation models to define the spatial extents influencing fluvial and estuarine catchments (J. Barton, Pope, Quinn, & Sherwood, 2008; Figure S1). In some instances, seagrass locations sampled were beyond fluvial and estuarine catchments defined, thus we allocated characteristics of the nearest catchment region to characterize catchment influences at these locations.

Plant community was defined in two ways. First, more generally as 'ecosystem' (mangrove forest, tidal marsh, or seagrass meadow) based on the plant cover where the sample was taken. Second, plant communities were further defined by either dominant species (for seagrasses, for which most were monotypic beds) or ecological vegetation class (EVC; for tidal marshes). Dominant species/EVC were determined for each sampling location based on % cover of 1-m$^2$ quadrat photos taken during sample collection. Tidal marsh EVCs sampled included coastal tussock saltmarsh, wet saltmarsh herbland, and wet saltmarsh shrubland, as described by Boon et al., 2011. Only one mangrove species is present in Victoria (the grey mangrove, *Avicennia marina*), therefore further classification of this ecosystem was not used. Seagrass species sampled included *Lepilaena marina*, *Posidonia australis*, *Ruppia megacarpa*, *Zostera muelleri*, and *Zostera nigricaulis*.

Topographical variables for each sample location included elevation and slope. Elevation data were obtained from the Victorian Coastal Digital Elevation Model 2017 (VCDEM 2017) from the Cooperative Research Centre for Spatial Information. Elevation data at 2.5 m spatial resolution were used where available. Where not available (for 2.8% of cores), 10 m spatial resolution elevation data were used to fill in the gaps. Slope was calculated from these data using the Slope tool in ArcMap (v. 10.2.2 for desktop). The elevation data are a composite product that integrated terrestrial and bathymetric LIDAR as well as multibeam sonar data. The vertical accuracies of the data varied with sensor setup for acquisition: ±10 cm at 1 sigma (68% conf. level) in bare ground for topographic LIDAR data (for the majority of our dataset), ±50 cm for bathymetric LIDAR, and ±<10 cm for multi-beam sonar data. Examples of spatial data used to develop models can be seen in Figure 2.Geomorphological setting was represented for each sample location using two proxies: distance to coast and distance to freshwater channel. For each, continuous Euclidean distance rasters at 10 m resolution were created for the feature of interest using the Euclidean Distance tool in ArcMap. Coastline and freshwater channel data came from the State of Victoria, Department of Environment, Land, Water & Planning 2018 (Victorian Coastline 2008 and Vicmap Hydro shapefiles, respectively). The Extract Values to Points tool in ArcMap was used to extract raster values to each sample location.

Primary lithology (rock type, i.e. potential sediment parent material) was defined as the rock type covering the greatest proportion of catchment area intersecting with sample locations. To calculate area of each lithology, the Tabulate Area tool was used in ArcMap based on the catchment region polygons. From the total area of each lithology in each catchment, the one with the greatest proportion was identified and input into a new field from which a new primary lithology raster was created. The Extract Values to Points tool in ArcMap was used to extract primary lithology raster values to each sample location. In total, 21 lithologies were identified in the dataset, 17 of which were identified as primary lithologies of the coastal catchments (Table S2).

Variables to assess the influence of anthropogenic processes on 30-cm deep sediment blue Cstocks included three relating to land use and one relating to human population. Primary land use for the catchment was

first defined as the primary land use (based on land use in individual polygons) covering the greatest proportion of catchment area. Land use spatial data was obtained from the Victorian Land Use Information System (2014/2015) from the Victoria State Government, Department of Economic Development, Jobs, Transport and Resources 2018. In total, nine general primary land use categories were identified in the dataset, all of which were identified as primary land uses of the coastal catchments (Table S3). The nine land use categories were pooled into three simplified categories: urbanized, agricultural, and natural. Then the areas of each within the catchment were summed and divided by total catchment area to provide the proportion of each catchment associated with those categories.

Human population densities were calculated for each catchment based on 2011 Australian census data, which were the most recent data available (Table S1). Population density was calculated for each district by dividing the population of the district by the area; this was then converted to a raster (100 $m^2$ resolution) to calculate the mean population density for the area of each catchment.

Complete details of data availability for inputs and outputs of our models can be found in supplementary Table S10.

### 2.3 Model generation, selection, averaging, and validation

To identify drivers of 30-cm deep sediment C stock variability and create the best predictive model of sediment C stocks to 30 cm deep we utilized a multi-step process based on an information theoretic approach and multimodel inference (Figure 3). Traditional approaches have relied on identification of the "best" data-based model; however, information-theoretic approaches allow for more reliable predictions through utilization of multiple models, especially in cases where lower ranked models may be essentially as good as the "best" ranked model (Burnham and Anderson, 2002; Symonds and Moussalli, 2011). Further, information theoretic model selection has been demonstrated to provide significant advantages for explaining phenomena with more complex drivers (Richards et al., 2011). Here, we first looked broadly at our variables of interest by narrowing down to the best models containing all possible variables ("global" models, as explained below) using $AIC_C$ (Akaike information criteria, corrected for small sample size) to explain the variability observed in the training dataset (70% of total C stock data; Symonds and Moussalli, 2011). From there, we identified which variables within the best global models best explained the observed variability in C stock data in order to remove unnecessary variables from the model equation (through the process of "dredging" and selecting the best subset, explained in detail below). The validity of removing unnecessary variables from the model is supported by the concept of parsimony, which suggests models more complicated than the best model provide little benefit and should be eliminated (Burnham and Anderson, 2002; Richards, 2008). The best subset of models generated from the global models ("dredge products") were selected based on delta $AIC_C$ <2, which are viewed as essentially interchangeable with the best model (Symonds and Moussalli, 2011). Each subset of best models was used to generate an averaged model, which was tested by generating predictions of C stocks for a reserved (30%) subset of the dataset. The best performing model was used to generate a predictive map of C stocks to 30 cm deep for mapped blue C ecosystems in Victoria. R code for this project can be found in the Harvard Dataverse (Ewers Lewis, 2020c).

To begin this process, potential ecological, geomorphological, and anthropogenic drivers were identified from the literature and relevant proxies were extracted from available spatial data using ArcMap (Table 1; Table S1). Predictor variable values derived from spatial data (along with our response variable values of C stocks) were

compiled into a master data table in ArcMap. Sample rows were randomly assigned as either "training" data to build the model (70% of the data) or "evaluating" data with which to validate the model (the remaining 30% of the data). The training dataset was imported into R (R Core Team, 2018) for further analysis.

Covariates were tested for correlation before composing the global models. From our 11 covariates of interest, covariate pairs were considered correlated and not used together in modelling based on a threshold value of ~≥0.4 correlation. The exception to this was covariate pairs that had a correlation value <0.4 but were still considered correlated by definition and therefore were not used together in modelling (e.g. proportion of catchment area urbanized and proportion agricultural, Figure S2). This resulted in four variables that did not correlate with other covariates and could be used together in all models (slope, distance to coast, distance to freshwater, and primary lithology – hereafter referred to as 'geomorphological covariates'), along with correlating covariates that fell into one of two groupings: 1) ecosystem, dominant species/EVC, and elevation were correlated (hereafter referred to as 'ecological covariates'; and 2) mean population density, proportion urbanized, proportion agricultural, and proportion natural land use were correlated (hereafter referred to as 'anthropogenic covariates').

As a first step, we aimed to identify which models that included all (non-correlated) variables were best for explaining the variability in C stock data. "Global" models (i.e. containing all possible variables) were created and ranked to identify the most important drivers of C stock variability. General linear mixed-effects models (GLMMs) were generated (family = gamma because our data were right-skewed; 'lme4' package v. 1.1-17; Bates et al. 2015) using all geomorphological covariates, along with one covariate each from the ecological and anthropogenic variable groups, resulting in 12 global models containing 6 covariates each (Table S4). Continuous covariates were scaled in R. Site (i.e. a single sampling area that contained from one to all three ecosystems) was used as a random effect in all models to account for spatial autocorrelation observed at ~78 km.

The 12 global models were ranked using $AIC_C$ ('AICcmodavg' package v. 2.1-1; Mazerolle, 2017; Table S5). The four best global models were chosen for further analysis based on delta $AIC_C$ ≤ ~5.0 compared to >30 for all other models. Because the top four global models all used dominant species/EVC as the ecological variable, this process was repeated for the next four best models – those that included "ecosystem" as the ecological predictor – to create averaged models that could be tested and used for predictions when more specific, spatially-explicit plant community data (i.e. dominant species/EVC) were not available.

The eight global models were "dredged" ('MuMIn' package v. 1.42.1; Barton, 2018) to assess the relative importance of covariates included in each model. In this context, "dredging" refers to the generation of a set of models that includes all possible combinations of fixed effects from the global model, containing from six to one variables (i.e. all combinations of five variables, all combinations of four variables, and so on). The dredge products of each global model (i.e. models created from "dredging") were ranked using $AIC_C$ and the best models (delta $AIC_C$ <2) were used to produce averaged models (named based on the global model they were generated from, e.g. global model 7 -> dredged and averaged -> averaged model 7). Averaged models were produced using the model.avg function ('MuMIn' package v. 1.42.1; Barton, 2018). The parameter estimates for each averaged model represent the average of that parameter's values from the models in which the variable appeared (from within the subset $AIC_C$ <2).

Averaged models were validated using the 30% evaluation dataset. Due to the limitations of using cross validation and bootstrapping on models with random effects (Colby and Bair, 2013), a direct comparison was done between predicted and actual values of the reserved dataset. The predict function in R was used to generate

predicted C stock values for 30-cm deep sediments using each of the eight averaged models on the reserved dataset. Each set of predicted values was compared to measured 30-cm deep sediment C stock values using a linear model to compute R-squared (adjusted) values. The models with the highest R-sq (adj) value from each set (one for "ecosystem" based models and one for "dominant species/EVC" based models) were applied to generate C stock predictions.

To test for differences in 30-cm deep sediment C stocks among species and EVCs, C stocks were log transformed to meet assumptions of normality and equal variances (log(Mg C ha$^{-1}$)) and a one-way analysis of variance (ANOVA) was run using dominant species/EVC as the factor. A Tukey's post-hoc analysis was used to distinguish groupings.

### 2.4 Prediction of 30-cm deep sediment blue C stocks

Spatial data relevant to the best ecosystem model were compiled for prediction of current ecosystem extent 30-cm deep sediment C stocks, and included rasters for total current ecosystem extent across Victoria (all mapped tidal marsh, mangrove, and seagrass), Euclidean distance to coast, and slope. Details and source information for all spatial data can be found in Table S1. All rasters were 10 m resolution and cut to the same extent using the Extract by Mask tool in ArcMap. The rasters were brought into R and processed using the raster package (Hijmans, 2017). Continuous variables were scaled to match the scaled variables of the model. Rasters were then compiled into a list, stacked, and used to generate a predictive raster map (*.tif file) of 30-cm deep sediment C stocks using the predict function. The C stock prediction raster (10 m resolution) was brought into ArcMap and resampled to 5 m resolution to better align to ecosystem extents. Sediment C stock values for each ecosystem extent were extracted to separate rasters and used to generate zonal statistics tables for estimating 30-cm deep sediment C stock sums and means. Rasters used for calculating C sums were converted to proper units to match map resolution using the Map Algebra tool (e.g. Mg C ha$^{-1}$ converted to Mg C per 25 m$^2$ raster cell). Sediment C stocks to 30 cm deep were summed for each ecosystem by catchment region, regions of interest, and the entire state. Regions of interest were identified visually as bays or estuaries hosting a substantial fraction of the state's blue C ecosystem distribution. Mapped predictions of modelled 30-cm deep sediment C stocks for this study can be found on the Harvard Dataverse (Ewers Lewis, 2020b).

### 3 Results

### 3.1 Drivers of 30-cm deep sediment blue C stock variability

Ranking of the 12 global models using AIC$_C$ suggested the ecological variable was the most important for determining model quality (Table S4 and S5). The top four models all contained dominant species/EVC as the ecological variable, with the following four containing ecosystem, and the remaining four containing elevation. The top four models fell within a delta AIC$_C$ value of ~5.0 and under, compared to the remaining models having delta AIC$_C$ values of ~35 or more, suggesting the top four models using dominant species/EVC were much better at explaining 30-cm deep sediment C stock variability than the remaining models. Within rankings for each ecological variable, anthropogenic variables in the top eight models ranked as follows, from highest to lowest importance: proportion catchment land use that is natural, proportion urbanized, mean population density, and proportion agricultural.

Dredging the top four global models and averaging the best dredge products (delta AIC$_C$ <2; Table S6) resulted in only three unique sets of model-averaged parameters (Table 2; full output can be seen in Table S7). The anthropogenic variables of mean population density and proportion agricultural land use did not appear in the best models produced from dredging global models 2 and 8, respectively. Therefore, both resulted in averaged models containing the same ecological and geomorphological variables, with no anthropogenic variable, and will hereafter be referred to as averaged model 2.

Parameter estimates from averaged models suggests dominant species/EVC was the most important predictor of 30-cm deep sediment C stocks, and was the only variable for which the 95% confidence interval of the estimates did not cross zero (Tables 2 and S7), suggesting a true effect of the variable on observed C stock variability (an estimate that included zero means there is potentially no impact of the variable on C stocks). Specifically, seagrasses *P. australis*, *R. megacarpa*, *Z. muelleri*, and *Z. nigricaulis* had 30-cm deep sediment C stocks significantly different than those of coastal tussock saltmarsh (assigned as the intercept in the model, or baseline dominant species/EVC for which to compare the effect of other dominant species/EVCs on C stocks), while all other tidal marsh EVCs, mangroves, and seagrass *L. marina* did not. This was confirmed by the ANOVA and Tukey's pairwise comparisons; there was a significant difference in 30-cm deep sediment C stocks based on dominant species/EVC ($F_{8,284}$ = 34.80, $p$ < 0.001, R-sq(adj) = 48.77 %); tidal marsh, mangrove, and seagrass *L. marina* had significantly higher C stocks than seagrasses *P. australis, Z. nigricaulis,* and *Z. muelleri* (Figure 4).

Across all three dominant species/EVC averaged models, distance to coast was the next most important geomorphological predictor, ranging from 50-51% relative importance compared to dominant species/EVC, followed by distance to freshwater (23-29% relative importance to dominant species/EVC), then slope (19-24% relative importance to dominant species/EVC). Of the two anthropogenic variables included, proportion urbanized land use was 47% relative importance compared to dominant species/EVC (averaged model 5) and proportion natural land use was 21% relative importance compared to dominant species/EVC (averaged model 11), suggesting proportion urbanized better explains variability in 30-cm deep sediment C stocks. The factor lithology did not appear in any of the best dredged models from the four global models.

For the next four averaged models, the ecological variable, ecosystem, was again the most important covariate (relative importance = 1.00; Table 3; Tables S8 and S9). Seagrasses impacted 30-cm deep sediment C stocks differently than tidal marshes (the intercept), as evidenced by the seagrass confidence intervals not crossing zero, while mangroves were no different than tidal marshes. However, in these averaged models, anthropogenic variables had greater relative importance than geomorphological predictors, unlike the models using dominant species/EVC as the ecological covariate. Proportion urbanization was still the most important anthropogenic variable, followed by proportion natural, but both had much higher relative importance (0.87 and 0.82, respectively) to the ecological variable compared to in the dominant species/EVC models. Additionally, mean population density appeared in one of the averaged models, though it did not appear in any of the dominant species/EVC models. Geomorphological variables, on the other hand, appeared less important in the ecosystem models than the dominant species/EVC models. Relative importance of distance to coast and slope were both lower than in the previous models, and distance to freshwater channels did not appear in the top dredged models with ecosystem at all.

**3.2 Model validation**

Comparison of 30-cm deep sediment C stock predictions from averaged models to actual C stock values in the 30% evaluation dataset show that our models accounted for ~44-49% of the observed variability in 30-cm deep sediment C stock values (Figure S3). The best three averaged models, using dominant species/EVC as the ecological predictor (averaged models 11, 5, and 2), had very similar R-sq(adj) values (ranging 0.4829-0.4881), with the best model (averaged model 2) being the one that did not include any anthropogenic variables. The same was true when comparing models using ecosystem as the ecological variable (averaged models 10, 4, 1, and 7) – the best R-sq(adj) was for the model with no anthropogenic variable (averaged model 7; 0.4618 compared to 0.4514, 0.4465, and 0.4566; Figure S3).

### 3.3 Modelled 30-cm deep sediment blue C stocks

We estimated a total of over 2.31 million Mg C stored in the top 30 cm of sediments in the ~68,700 ha of blue C ecosystems across Victoria (Table 4; Figure 5). This estimate is based on predictions from our best averaged model that utilized ecosystem type as the ecological variable (averaged model 7), which explained 46.18% of observed variability in C stock data and had an RSE of 39.29. Tidal marshes stored 48.2 %, mangroves stored 11.0 %, and seagrasses stored 40.8 % of total predicted 30-cm deep sediment C stocks. Mean predicted sediment C stocks (±SD) to 30 cm deep for each ecosystem type were 57.96 (±2.90) Mg C ha$^{-1}$ for tidal marsh, 50.64 (±1.35) Mg C ha$^{-1}$ to mangroves, and 23.48 (±0.57) Mg C ha$^{-1}$ for seagrass based on predicted C stock values in all raster cells of each ecosystem's mapped areal extent in Victoria. These 30-cm deep sediment C stock values ranged from 23.33 – 291.18, 23.34 – 77.81, and 23.33 – 73.42 Mg C ha$^{-1}$ for tidal marsh, mangroves, and seagrass, respectively.

Fourteen areas of the coast were identified as regions of interest (ROIs) and contained over 99.5% of Victoria's total 30-cm deep sediment blue C stocks (Table 5) in 95.6% of the state's blue C ecosystem area (~65,700 ha). Of these regions, four of them contained over 87.6% of total estimated 30-cm deep sediment C stocks in 86.5% (~59,410 ha) of the state's blue C ecosystem area. Listed from highest to lowest C stocks, they were: Corner Inlet, Westernport Bay, Gippsland Lakes, and Port Phillip Bay.

### 4 Discussion

### 4.1 Drivers of 30-cm deep sediment blue C stock variability

Our best model explained 48.8% of the observed variability in 30-cm deep sediment C stocks, with the ecological variable, i.e. plant community, being the greatest predictor of C stock variability in all of the models. Plant community is related to C stocks both directly and indirectly through correlation with other variables driving C stock variability. Plant morphology may directly influence 30-cm deep sediment C stocks through the magnitude of plant biomass contributed to autochthonous C stocks and through an interaction with hydrodynamics. For example, higher C stock values in larger seagrass species, such as *P. australis*, are thought to be linked to both higher inputs of autochthonous C (larger rhizomes with more refractory C), and better particle trapping via a deeper canopy, which reduces water velocities and resuspension (Lavery et al., 2013). Under similar hydroperiods, saltmarsh grasses have been shown to have better sediment trapping abilities compared to mangrove trees (Chen et al., 2018), further suggesting plant traits (e.g. productivity and morphology) are an important driver of C stocks, rather than indirect impacts of inundation regimes alone.

Plant community is correlated with a number of other variables that may influence C storage, such as inundation regimes. Within and among similar ecosystems, elevation is a proxy for inundation regimes and can drive differences in C stocks. For example, in southeast Australia, tidal marshes in the upper intertidal zone had lower C accumulation rates than mangroves, with the cause hypothesized to be that the tidal inundation was shallower, less frequent, and for shorter durations, limiting the amount of allochthonous C accumulation (Saintilan

et al., 2013). This appeared to be a more important driver in C accumulation variability than the difference in biomass production between the two ecosystems (Saintilan et al., 2013), highlighting the importance of elevation in determining C stocks. In our study, elevation was correlated to ecosystem and dominant species/EVC, so the differing effects of elevation compared to vegetation community could not be teased apart without violating assumptions of non-collinearity in our models. However, the higher ranking of global models with dominant

species/EVC or ecosystem above those with elevation in our study suggests plant community itself is a better predictor of 30-cm deep sediment C stocks than simply position in the tidal frame.

       Our global models specifying dominant species (for seagrass meadows) or EVC (for tidal marshes) ranked higher in our model selection than those that only specified the ecosystem (i.e. tidal marsh, mangrove, or seagrass). This ranking was supported by our model validation, in which our averaged model that best explained

30-cm deep sediment C stock variability included dominant species/EVC and accounted for 48.8% of the variability observed (Figure S3). Still, the best averaged model containing ecosystem as the ecological predictor performed nearly as well, and explained 46.2% of the variability. These results suggest that even when specific data on species composition are not available, 30-cm deep sediment C stocks can be estimated with a similar degree of confidence based on ecosystem type, which is often a much more readily available form of data and

therefore favorable for calculating sediment C stocks in data-deficient areas.

       Geomorphological variables were more important than most anthropogenic variables in our models (Tables 2 and 3). Though lithology was not part of our averaged models, it is possible that its exclusion was due mostly to scale (catchment) and it may be important when accounted for on a more local scale. Distance to coast, distance to freshwater channels, and slope all appeared in the averaged models using dominant species/EVC, with

distance to coast being most important. However, in models using ecosystem, distance to freshwater channels was no longer important enough to appear in the averaged models, and the anthropogenic variables, proportion urbanized and proportion natural, were more important than any of the geomorphological variables. Model validation revealed that the best predictions for either set of models (those using dominant species/EVC and those using ecosystem as the ecological variable) came from the model that did not include any anthropogenic variables.

Although our models suggest anthropogenic variables have little impact on 30-cm deep sediment C stocks, it is more likely that anthropogenic variables are impacting processes we could not measure. For example, excess nutrients resulting from certain land uses may stress plants to the point of affecting survival and therefore sediment stability (Macreadie et al., 2017b); without measuring changes to ecosystem distribution or sediment thickness (i.e. erosion) we could not pick up on these sediment C losses. Similarly, though enhanced sedimentation

rates may increase C burial in catchments with certain land uses (e.g. high population density or high area of agriculture; Yang, Kanae, Oki, Koike, & Musiake, 2003), this addition to C stocks would be reflected in sequestration rate, which we did not measure in this study.

       Proxies for the drivers of sediment C stock variability can be quantified and described for modelling in numerous ways. Though we maximized our ability to choose variables representing meaningful relationships with

30-cm deep sediment C stocks by alternating the forms of the anthropogenic variables tested in our models (i.e. proportion urban vs. proportion agriculture vs. proportion natural v. mean population density), it may be beneficial to incorporate more direct measures of anthropogenic impacts in C stock modeling, such as nutrients and suspended particulate organic matter coming from catchments.

We also aimed to maximize our ability to capture relationships between contemporary drivers and
sediment C stocks by utilizing sediment C stock data to only 30 cm deep, a sediment horizon more directly impacted by recent environmental conditions compared to deeper stocks due to age. Based on previously estimated sediment accretion rates in blue C ecosystems in the study region (averaging 2.51 to 2.66 mm year$^{-1}$ in tidal marshes (Ewers Lewis et al., 2019; Rogers et al., 2006a) and 7.14 mm year$^{-1}$ in mangroves (Rogers et al., 2006a)), the top 30 cm of sediment represents roughly ~113-120 years of accretion in Victorian tidal marshes and ~42
years of accretion in Victorian mangroves. These time scales suggest sediments depths utilized in this study are more appropriate for assessing the impacts of modern environmental conditions on sediment C stocks compared to meter-deep stocks, which can be thousands of years old (e.g. Ewers Lewis et al., 2019). Using 30-cm deep sediment C stocks also allows us to be more confident that the vegetation present now has been there during the time of sediment accretion, unlike deeper sediments that are thousands of years old and for which it is difficult to
determine what vegetation, if any, was present at the time of accretion.

The variability in 30-cm deep sediment C stocks that could not be explained by our modeling may also be related to the inherent challenges surrounding spatial and temporal matching of driver proxies and sediment C stock measurements; the relationship between 30-cm deep sediment C stocks and contemporary environmental settings can be represented more accurately for some variables over others.

Ecosystem type was a relatively powerful predictor of 30-cm deep sediment C stock variability in our study and this is likely due, in part, to the direct relationship between vegetation and surface sediments. In most vegetated ecosystems, the majority of underground plant biomass and microbial activity exists within the top 20 cm of soils (Trumbore, 2009). For saltmarsh, it has been demonstrated that the top 30 cm of sediment are directly impacted by current vegetation (Owers et al., 2016). Therefore, using 30-cm deep sediment C stock measurements
allowed us to target the portion of the sediment profile most likely to be influenced by current vegetation.

The portion of recently accreted sediments influenced by contemporary anthropogenic drivers is harder to identify than that influenced by ecosystem vegetation. Based on estimated accretion rates for this region from the literature (Ewers Lewis et al., 2019; Rogers et al., 2006b), 30-cm deep sediments would have taken an average of ~80 years to accumulate in Victoria (~117 years in tidal marsh and ~42 years in mangroves). Though
sedimentation rates vary over time, they are relatively steady in comparison to changes in anthropogenic drivers, such as land use change, which can happen abruptly. This means that modern day maps of land use, though useful for looking at the general impact of human activities on ecosystem processes, may be more useful for relating to variability in sediment C stocks when the data is assessed at finer temporal resolutions. For example, comparing land use area data across various time periods with C densities in aged bands of sediment could help capture the
pulse effects of sudden land use changes in narrower sediment horizons representative of the same time periods. In this study, the effects of land-use change may have been too diluted within the 30-cm horizons to relate to impacts on sediment C stock.

Spatially, anthropogenic variables are also difficult to assign to particular ecosystem locations or depths. Many blue C ecosystems in Victoria are located in coastal embayments and receive inputs from multiple

catchments, making the influence of specific areas of land-use or population changes difficult to track to specific ecosystem locations. Modern-day factors influencing vegetation can also have impacts on C stocks deeper than the sediments we measured. The effects of underground biomass on sediment C stocks can extend beyond the top 30 cm, and in fact new C inputs and active C cycling by microbial communities can occur as deep as underground roots extend (Trumbore, 2009). These new C additions (and fluxes) at depth fall outside the general pattern of

sediment C decay down-core in vegetated ecosystems (Trumbore, 2009) which has previously allowed for linear or logarithmic regressions to be used to extrapolate 1-m deep C contents from shallow (e.g. 30-50 cm deep) sediment C data (Macreadie et al., 2017a; Serrano et al., 2019). The activity of underground biomass and microbes at depth, when considered over space and time, may account for large C fluxes. The influence of anthropogenic activities, such as land use changes, on these processes via impacts to vegetation may largely go unnoticed based

on current methods (Trumbore, 2009), both in this study and in blue C stock assessments on larger scales. We suggest further research to understand the dynamics of active C cycling at sediment depths traditionally considered stable.

Another limitation to C stock modelling is knowledge of environmental features that may be important in influencing C storage, but are generally not monitored. For example, the maturity of a blue C ecosystem can

affect C storage and composition (Kelleway et al., 2015). Within a single saltmarsh species, the maturity of the system is a major factor determining the role of the marsh as a C sink. Mature systems of *Spartina maritime* have higher C retention—via higher belowground production, slower decomposition rate, and higher C content in sediments—than younger *S. maritime* marsh systems (Sousa et al. 2010b). Mature marshes have also been observed to have greater contributions of allochthonous C storage over time, while younger marshes

predominantly have autochthonous organic matter signatures (Chen et al. 2015, Tu et al. 2015). Long-term mapping of blue C ecosystems could be beneficial for tracking maturity of vegetation for C stock modelling, as well as reduce the error in C stock measurements associated with changes to blue C ecosystem area.

Finally, we suggest future studies examine the relationship between the drivers we have described and individual blue C ecosystem types in order to further refine sediment blue C stock modelling. With a large dataset

from a single ecosystem, relationships may be identified that were overshadowed in this study by the inclusion of all three ecosystems. For example, because elevation correlated with our two ecological variables it was not included in our best models. However, within a single ecosystem, elevation may be an important driver of sediment C stock variability due to its relationship with inundation regimes (Chen et al., 2015; Chmura et al., 2003; Chmura and Hung, 2004).


**4.2 Modelled 30-cm deep sediment blue C stocks**

Our estimate of 2.31 million Mg C stored in the top 30 cm of sediment in all blue C ecosystems in Victoria was about 20% lower than that of Ewers Lewis et al. (2018), who estimated 2.91 million Mg C based on the same C stock data, but calculated total stocks based on average C stock values and ecosystem extent in each

of the five coastal catchments. These results suggest that modelling 30-cm deep sediment C stocks based on environmental drivers may reduce the chances of overestimating sediment C stocks by better accounting for fine-scale variability. Our modelled 30-cm deep sediment C stock estimates support our earlier findings that tidal marshes store more C than any other blue C ecosystem in Victoria. Our estimates are now refined in that modelled

stocks suggest tidal marshes store closer to 48% (rather than 53%) and seagrasses store closer to 41% (rather than 36%) of total 30-cm deep sediment blue C stocks (Ewers Lewis et al., 2018). Our original estimate of mangrove contribution to total blue C was supported by our modelling – by either method we estimated mangroves to store 11% of Victoria's 30-cm deep sediment blue C stocks.

It is important to emphasize here that total sediment depths in blue C ecosystems can vary greatly, and are commonly deeper than 30 cm. Blue C ecosystems can have sediments up to several meters deep (e.g. Lavery et al., 2013; Scott and Greenberg, 1983), suggesting the estimates of C stocks measured here are conservative. In spite of these limitations, 30-cm deep sediment C stock estimates give us valuable knowledge about the sediment C pool most vulnerable to disturbance and how it may be impacted by environmental drivers.

In examining C stocks within ROIs, i.e. areas of the coast containing substantial distributions of blue C ecosystems, we found that just four of the 14 ROIs housed nearly 88% of 30-cm deep sediment blue C stocks in the state, a direct reflection of the large proportion of blue C ecosystem area in these regions (nearly 87% of the state's total blue C area). This trend appears to be driven by the presence of extensive seagrass sediment C stocks (Table 5) in these four regions, accompanied by extensive tidal marsh sediment C stocks. This result has important implications for management of coastal blue C. In cases where resources are limited, identification of areas housing major blue C sinks, in conjunction with evaluation of other ecosystem services, can help provide insight to guide conservation strategies. For example, strategies to conserve tidal marshes in the four major ROIs could serve the additional purpose of helping to preserve the adjacent seagrass meadows via facilitation; tidal marshes serve as filters of excess nutrients coming down from the catchment (Nelson and Zavaleta, 2012) that may otherwise cause a loss of seagrass beds due to light reduction resulting from the growth of algal epiphytes, macroalgae, and phytoplankton (Burkholder et al., 2007). Further, our mapping of within-ecosystem-patch variability in 30-cm deep sediment C stocks is an important output for facilitating management actions on an applicable level, allowing priority of particular parts of an ecosystem patch for conservation when necessary.

**5 Conclusions**

In this study, we had the unique opportunity to assess a large regional dataset of 30-cm deep sediment blue C stocks to explore the influence of ecological, geomorphological, and anthropogenic variables in driving sediment blue C stock variability. Because of the high spatial resolution of sampling within similar latitudes we were able to focus on variables driving differences in 30-cm deep sediment C stocks within catchments. We found that plant community was most important for determining 30-cm deep sediment C stocks and that combining this variable with geomorphological variables relating to position in the catchment allowed us to model stocks at a fine spatial resolution. Identification and mapping of these dense 30-cm deep sediment blue C sinks in Victoria, in conjunction with evaluation of other ecosystem services, will be useful for conservation management regionally, for example through the identification of hotspots for protection and key locations for restoration efforts. We recommend these methods be tested in other areas of the globe to determine whether they may be applicable for identifying relationships between potential environmental drivers and sediment blue C stocks and creating predictive sediment C stock models and maps for blue C ecosystems at scales relevant to resource management applications in other regions.

*Author Contributions*. CEL, DI, MY, and PM conceived the study. CEL, JB, BH, JS, PC, and PM produced the input carbon data for the model. CEL and MY wrote the code. CEL analyzed the data, performed the calculations, and produced the GIS data and maps. CEL prepared the paper with contributions from all authors.

*Competing Interests*. The authors declare that they have no conflict of interest.

*Data Availability.* The data associated with this study are accessible through the Harvard Dataverse:

Ewers Lewis, C. J.: R Code for 30-cm Sediment Blue Carbon Stock Modelling, doi:10.7910/DVN/0WKEHJ, 2020.

Ewers Lewis, C. J.: Model Predictions Map: 30-cm Deep Sediment Blue Carbon Stocks for Victoria, Australia, doi:10.7910/DVN/UDOAUT, 2020.

*Acknowledgements.* We thank Parks Victoria and the Victorian Coastal Catchment Management Authorities (CMAs) for their support and funding: Marty Gent & Glenelg Hopkins CMA, Chris Pitfield & Corangamite CMA, Emmaline Froggatt & Port Phillip Westernport CMA, Belinda Brennan & West Gippsland CMA, and Rex Candy & East Gippsland CMA. Funding was provided by an Australian Research Council DECRA Fellowship (DE130101084) and an Australian Research Council Linkage Project (LP160100242). CEL also thanks the University of Technology Sydney for scholarship support.

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

**Table 1.** Hypothesized drivers of 30-cm deep sediment blue C stock variability. Drivers were grouped into three categories: 1) ecological (ecosystem type and dominant species/ecological vegetation class), 2) geomorphological (elevation, slope, distance to freshwater channel, distance to coast, and lithology), and 3) anthropogenic (land use and population). A more detailed explanation of driver rationale, along with literature and spatial data references, can be found in Table S1.

| Driver | Hypothesis and rationale |
|---|---|
| *Ecological* | |
| **ECOSYSTEM TYPE** | Ecosystem is the dominant driver of C stock variability<br>➢ C stocks differ by ecosystem type due to: 1) differences in position in the tidal frame, and 2) differences in morphology, which influence settling and trapping of suspended particles, as well as production of autochthonous C inputs. |
| **DOMINANT SPECIES OR ECOLOGICAL VEGETATION CLASS** | Species composition better explains C stock variability than ecosystem alone<br>➢ C stocks vary across species and community composition, as well as elevations. |
| *Geomorphological* | |
| **ELEVATION** | Lower elevations are correlated with higher C stocks<br>➢ Lower elevations have higher sedimentation rates, aiding the trapping of organic C, and are inundated more often, providing more opportunity for contribution of allochthonous C. |
| **SLOPE** | Shallower slopes are correlated with higher C stocks<br>➢ Steeper slopes are more vulnerable to erosion and less conducive to sedimentation and particle trapping than shallower slopes. |
| **DISTANCE TO FRESHWATER CHANNEL** | Distance to freshwater channel is negatively correlated with C stocks<br>➢ Being in close proximity to freshwater inputs may increase plant growth via freshwater and nutrient inputs, and enhance C preservation through delivery of smaller grain size particles. |
| **DISTANCE TO COAST** | C stocks are greater higher up in the catchment<br>➢ Greater inputs of organic C from terrestrial sources higher in the catchment result in higher sediment C stocks. |
| **LITHOLOGY** | C stocks vary with terrestrial parent material of sediments<br>➢ Rock type may influence grain size and mineral content of sediments exported from catchments; smaller grain sizes and certain minerals enhance C stocks and preservation. |
| *Anthropogenic* | |
| **LAND USE** | C stocks vary based on land use activities in the catchment<br>➢ Export of terrestrial C, nutrients, and sediments varies by land use, especially when comparing urbanized, agricultural, and natural land uses. |
| **POPULATION DENSITY** | C stocks differ across population levels due to a correlation with land use<br>➢ Increases in population size lead to increases in urbanisation and competition for land use. |

**Table 2.** Parameter estimates for averaged models containing dominant species/ecological vegetation class (EVC) as the ecological variable. Parameter estimates were calculated based on averaging the best model products (delta $AIC_C$ <2) resulting from dredging the top four dominant species/EVC global models (global model 11, 5, 2, and 8; descriptions of global models can be found in Table S4). Note that averaged model 2 and 8 are the same because neither of the anthropogenic covariates from the global models (mean population density and proportion agricultural land use for global models 2 and 8, respectively) appeared in the best dredge model products. DSE = dominant species/EVC; DSE are color-coded by ecosystem type: red = tidal marsh, green = mangrove, blue = seagrass; Adj SE = adjust standard error; RI = relative importance. N/A = the parameter was not included in the averaged model.

| Parameter | Averaged Model 11 | | | | Averaged Model 5 | | | | Averaged Model 2 | | | |
|---|---|---|---|---|---|---|---|---|---|---|---|---|
| | Estimate | ± | Adj SE | RI | Estimate | ± | Adj SE | RI | Estimate | ± | Adj SE | RI |
| Intercept DSE: coastal tussock saltmarsh | 0.0177 | ± | 0.0043 | | 0.0171 | ± | 0.0042 | | 0.0176 | ± | 0.0042 | |
| DSE: wet saltmarsh herbland | 0.0012 | ± | 0.0041 | 1.00 | 0.0013 | ± | 0.0040 | 1.00 | 0.0011 | ± | 0.0041 | 1.00 |
| DSE: wet saltmarsh shrubland | -0.0027 | ± | 0.0042 | " | -0.0023 | ± | 0.0042 | " | -0.0028 | ± | 0.0042 | " |
| DSE: *A. marina* | 0.0011 | ± | 0.0041 | " | 0.0015 | ± | 0.0041 | " | 0.0011 | ± | 0.0041 | " |
| DSE: *L. marina* | -0.0024 | ± | 0.0051 | " | -0.0020 | ± | 0.0051 | " | -0.0024 | ± | 0.0051 | " |
| DSE: *P. australis* | 0.0394 | ± | 0.0179 | " | 0.0405 | ± | 0.0179 | " | 0.0412 | ± | 0.0180 | " |
| DSE: *R. megacarpa* | 0.0903 | ± | 0.0313 | " | 0.0908 | ± | 0.0314 | " | 0.0909 | ± | 0.0313 | " |
| DSE: *Z. muelleri* | 0.0291 | ± | 0.0047 | " | 0.0295 | ± | 0.0047 | " | 0.0292 | ± | 0.0047 | " |
| DSE: *Z. nigricaulis* | 0.0397 | ± | 0.0172 | " | 0.0389 | ± | 0.0172 | " | 0.0398 | ± | 0.0172 | " |
| Distance to coast | -0.0011 | ± | 0.0015 | 0.51 | -0.0011 | ± | 0.0015 | 0.51 | -0.0011 | ± | 0.0015 | 0.50 |
| Distance to freshwater | -0.0005 | ± | 0.0014 | 0.23 | -0.0006 | ± | 0.0015 | 0.29 | -0.0007 | ± | 0.0015 | 0.29 |
| Slope | -0.0001 | ± | 0.0004 | 0.19 | -0.0002 | ± | 0.0005 | 0.23 | -0.0002 | ± | 0.0005 | 0.24 |
| Proportion natural | 0.0003 | ± | 0.0009 | 0.21 | N/A | | N/A | N/A | N/A | | N/A | N/A |
| Proportion urbanized | N/A | | N/A | N/A | -0.0010 | ± | 0.0014 | 0.47 | N/A | | N/A | N/A |

**Table 3.** Parameter estimates for averaged models containing ecosystem as the ecological variable. Parameter estimates were calculated based on averaging the best model products (delta AIC$_C$ <2) resulting from dredging the four global models that used ecosystem as the ecological variable (global models 10, 4, 1, and 7; descriptions of global models can be found in Table S4), combined with geomorphological and anthropogenic variables as specified. Ecosystems are color-coded for consistency: red = tidal marsh, green = mangrove, blue = seagrass; Adj SE = adjust standard error; RI = relative importance. N/A = the parameter was not included in the averaged model.


| Parameter | Averaged Model 10 | | | | Averaged Model 4 | | | | Averaged Model 1 | | | | Averaged Model 7 | | | |
|---|---|---|---|---|---|---|---|---|---|---|---|---|---|---|---|---|
| | Estimate | ± | Adj SE | RI | Estimate | ± | Adj SE | RI | Estimate | ± | Adj SE | RI | Estimate | ± | Adj SE | RI |
| Intercept Ecosystem: tidal marsh | 0.0178 | ± | 0.0020 | | 0.0166 | ± | 0.0018 | | 0.0174 | ± | 0.0020 | | 0.0174 | ± | 0.0020 | |
| Ecosystem: mangrove | 0.0022 | ± | 0.0013 | 1.00 | 0.0024 | ± | 0.0013 | 1.00 | 0.0022 | ± | 0.0013 | 1.00 | 0.0022 | ± | 0.0013 | 1.00 |
| Ecosystem: seagrass | 0.0244 | ± | 0.0026 | " | 0.0254 | ± | 0.0025 | " | 0.0252 | ± | 0.0025 | " | 0.0252 | ± | 0.0025 | " |
| Distance to coast | -0.0009 | ± | 0.0014 | 0.45 | -0.0006 | ± | 0.0010 | 0.39 | -0.0003 | ± | 0.0008 | 0.22 | -0.0003 | ± | 0.0008 | 0.27 |
| Slope | -0.0002 | ± | 0.0006 | 0.30 | -0.0002 | ± | 0.0005 | 0.29 | -0.0002 | ± | 0.0005 | 0.21 | -0.0002 | ± | 0.0005 | 0.26 |
| Proportion natural | 0.0022 | ± | 0.0017 | 0.82 | N/A | ± | N/A | N/A | N/A | ± | N/A | N/A | N/A | ± | N/A | N/A |
| Proportion urbanized | N/A | ± | N/A | N/A | -0.0024 | ± | 0.0015 | 0.87 | N/A | ± | N/A | N/A | N/A | ± | N/A | N/A |
| Mean population density | N/A | ± | N/A | N/A | N/A | ± | N/A | N/A | -0.0001 | ± | 0.0004 | 0.18 | N/A | ± | N/A | N/A |

**Table 4.** Blue C ecosystem area (ha) and modelled 30-cm deep sediment C stocks (Mg C) by catchment region and total across the state (Victoria, Australia).

| Catchment Region | Tidal Marsh Area (ha) | Tidal Marsh C stocks (Mg C) | Mangrove Area (ha) | Mangrove C stocks (Mg C) | Seagrass Area (ha) | Seagrass C stocks (Mg C) | All Blue C Ecosystems in Victoria Total area (ha) | All Blue C Ecosystems in Victoria Total blue C stock (Mg C) |
|---|---|---|---|---|---|---|---|---|
| Glenelg Hopkins | 138 | 6,828 | 0 | N/A | 32 | N/A | 170 | 6,828 |
| Corangamite | 3,010 | 187,943 | 58 | 3,022 | 5,355 | 128,117 | 8,423 | 319,083 |
| Port Phillip & Westernport Bays | 3,108 | 158,604 | 1,828 | 90,359 | 14,457 | 328,725 | 19,393 | 577,688 |
| West Gippsland | 13,038 | 711,083 | 3,301 | 161,652 | 17,508 | 413,642 | 33,847 | 1,286,377 |
| East Gippsland | 1,332 | 50,504 | 0 | N/A | 5,552 | 72,873 | 6,884 | 123,377 |
| **Total** | **20,626** | **1,114,961** | **5,187** | **255,034** | **42,903** | **943,357** | **68,715** | **2,313,352** |


**Table 5.** Modelled 30-cm deep sediment blue C stocks (Mg C) by region of interest (ROI; listed from West to East). N/A = Ecosystem does not occur in ROI.

| Region of Interest | C stocks (Mg C) by Ecosystem | | | All Blue C Ecosystems in ROI |
|---|---|---|---|---|
| | Tidal Marsh | Mangrove | Seagrass | |
| Breamlea | 18,650 | N/A | N/A | 18,650 |
| Lake Connewarre/Barwon Heads | 101,218 | 2,890 | N/A | 104,109 |
| Port Phillip Bay | 105,169 | 243 | 156,824 | 262,236 |
| Westernport Bay | 120,827 | 90,248 | 300,420 | 511,495 |
| Andersons Inlet | 18,992 | 7,455 | 890 | 27,337 |
| Shallow Inlet | 9,384 | N/A | 19,778 | 29,162 |
| Corner Inlet | 253,367 | 154,198 | 346,317 | 753,882 |
| Jack Smith Lake | 73,839 | N/A | N/A | 73,839 |
| Lake Denison | 7,353 | N/A | N/A | 7,353 |
| Gippsland Lakes | 391,023 | N/A | 99,267 | 490,291 |
| Lake Corringle | 3,449 | N/A | N/A | 3,449 |
| Bemm River region | N/A | N/A | 7,806 | 7,806 |
| Tamboon Inlet | N/A | N/A | 2,563 | 2,563 |
| Wallagaraugh River/Mallacoota region | 3,180 | N/A | 8,117 | 11,296 |
| **Total** | **1,106,452** | **255,034** | **941,982** | **2,303,468** |

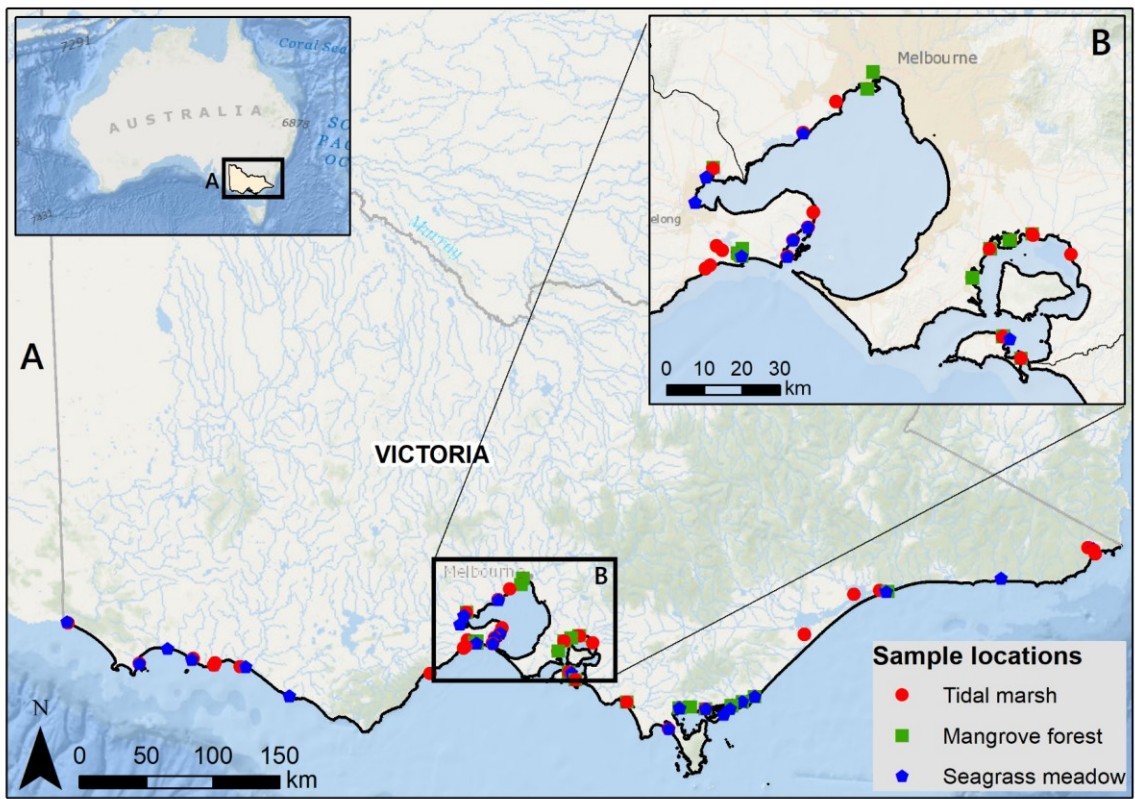

**Figure 1.** Sample locations for 30-cm deep sediment blue C stock measurements across Victoria, Australia (A), focusing in on Port Phillip and Westernport Bays (B). Service Layer Credits: Esri, Garmin, GEBCO, NOAA NGDC, and other contributors. Adapted from Ewers Lewis et al., 2018.

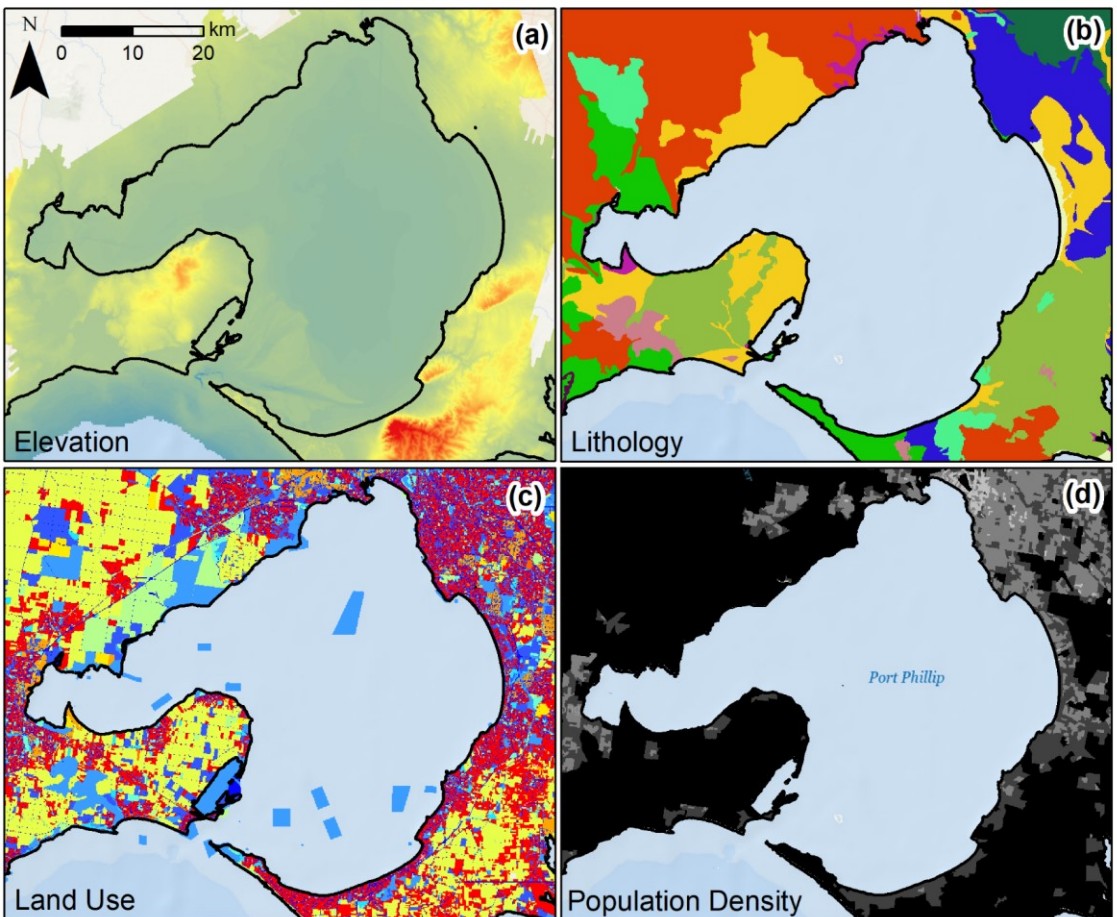

**Figure 2.** Variability of select potential C stock drivers in Port Phillip Bay, Victoria, Australia. Raw spatial data layers were processed to define covariate values at each sample location or for the catchment of the sample location. Pictured layers include: (a) elevation raster at 10 m resolution, (b) lithology polygons, (c) land use polygons, and (d) and population density polygons.

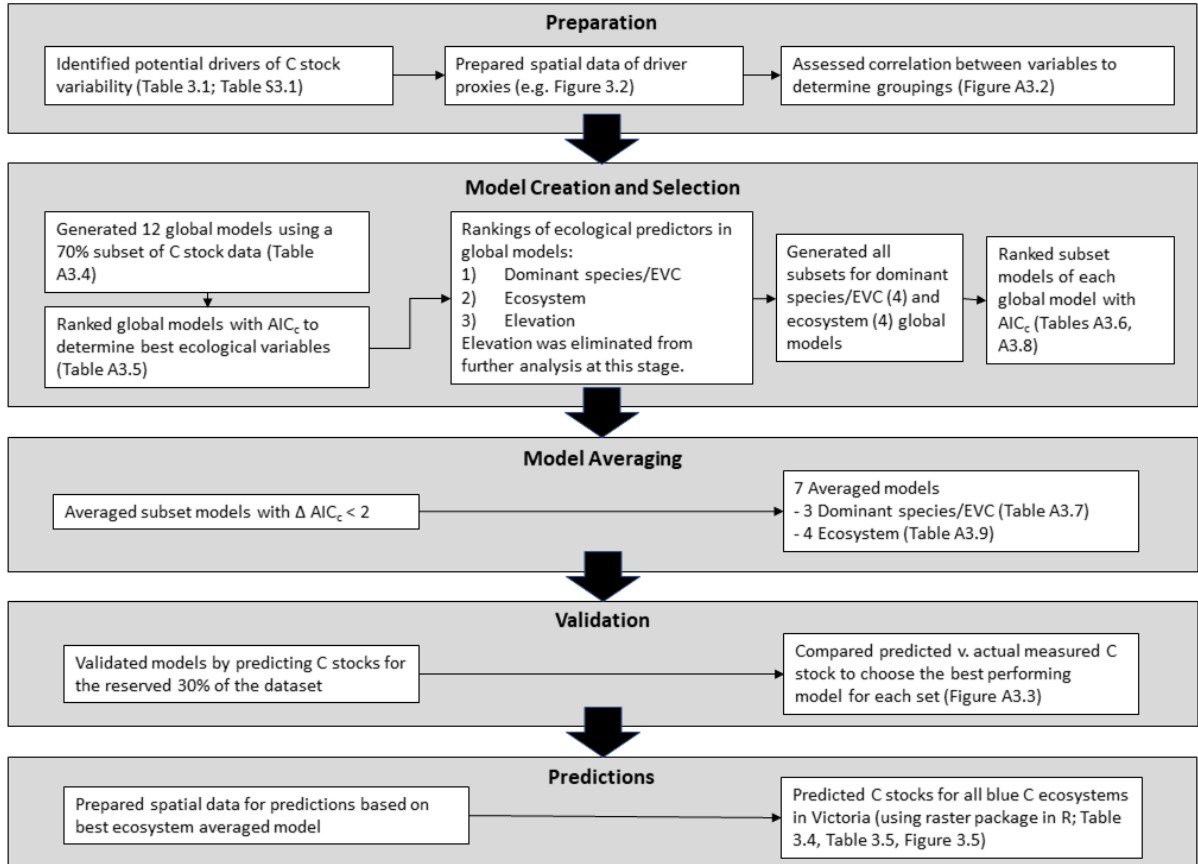

**Figure 3.** Conceptual workflow of sediment C stock modelling methods: preparation, model creation and selection, model averaging, validation, and predictions.

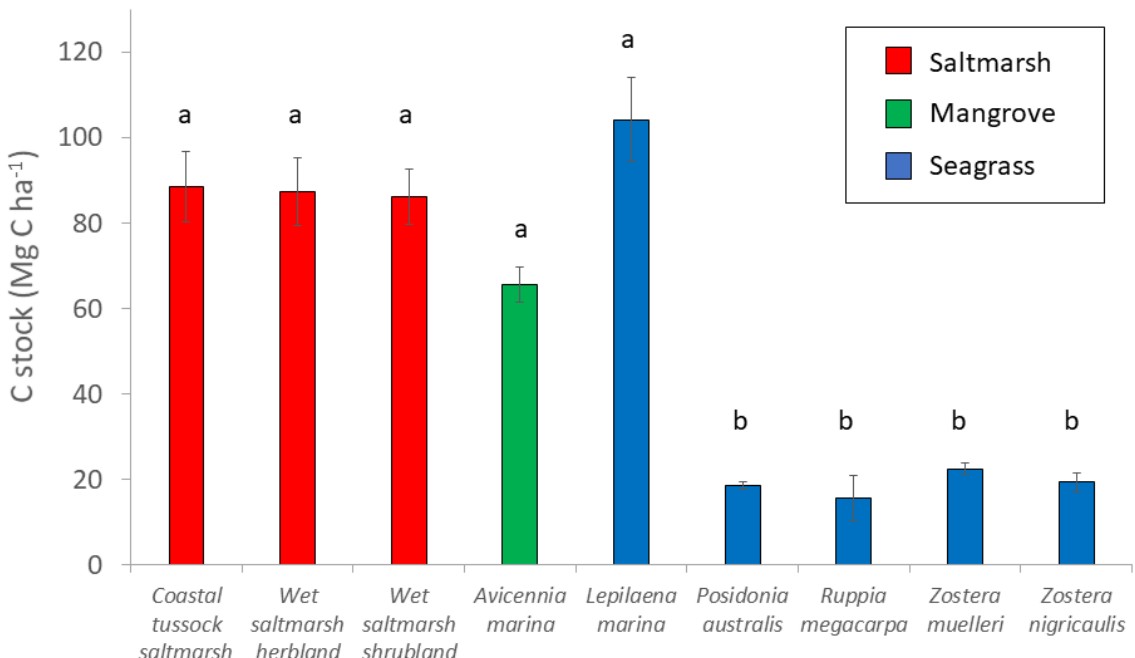

**Figure 4.** Measured C stocks (Mg C ha$^{-1}$; average ± standard error) in the top 30 cm of sediment by dominant species/ecological vegetation class (EVC). Bars are color-coded by ecosystem type: red = tidal marsh, green = mangrove, blue = seagrass. C stocks differed significantly by dominant species/EVC, with higher C stocks in coastal tussock saltmarsh, wet saltmarsh herbland, wet saltmarsh shrubland, mangroves *A. marina*, and seagrass *L. marina* (group a) compared to seagrasses *P. australis, Z. nigricaulis,* and *Z. muelleri* (group b; ANOVA and

Tukey pairwise comparison, $F_{8,284}$ = 34.80, $p < 0.001$, R-sq(adj) = 48.77 %). Error bars represent standard error of the mean.

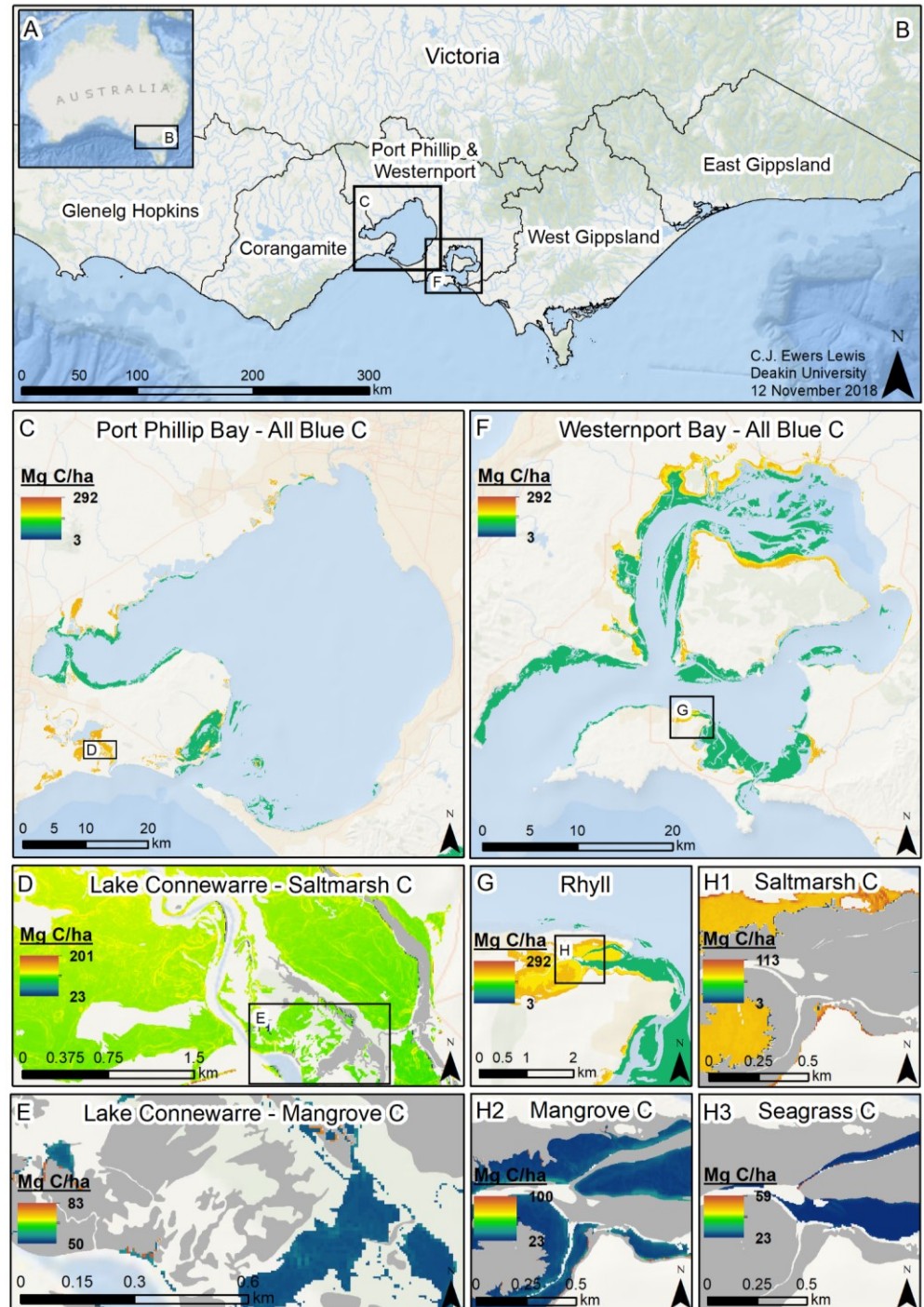

**Figure 5.** Modelled 30-cm deep sediment blue C stocks for Victoria, Australia. Location of Victoria in Australia (A), coastal catchment regions of Victoria (B), modelled C stocks for all blue C ecosystems in Port Phillip Bay (C), modelled saltmarsh C stocks in Lake Connewarre (D); modelled mangrove C stocks in subsection of Lake Connewarre (E);  modelled C stocks for all blue C ecosystems in Westernport Bay (F); modelled C stocks for all blue C ecosystems in Rhyll (Phillip Island) (G); and modelled saltmarsh C stocks (H1), mangrove C stocks (H2), and seagrass C stocks (H3) in subsection of Rhyll. Basemap service layer credits: Esri, Garmin, GEBCO, NOAA NGDC, and other contributors.