# Peer review of "Drivers and modelling of blue carbon stock variability in"

_Biogeosciences, 2019_

## Referee Comment (RC1) · Anonymous Referee #1 · 1 Oct 2019

Review of bg-2019-294 "Drivers and modelling of blue carbon stock variability" by Ewers Lewis et al. In this paper, the authors look to create a framework for modelling shallow carbon stocks (0-30cm) in vegetated coastal ecosystems. They use a combination of geomorphological, anthropogenic and ecological variables, combined with carbon stocks from a large number of shallow cores (n = 287) to construct the model and estimate carbon stocks for a region in southern Australia. The model could account for $\sim$ 49% of the variability in shallow carbon stocks, with plant community being the strongest predictor in the model.

Generally this is an interesting paper, but I have some points that should be clarified.
1. Can these shallow carbon stocks be considered "blue carbon"? There is growing evidence that carbon within the surface layers is still highly susceptible to degradation.

A typical profile of carbon down the soil profile in these vegetated habitats show a decline with depth until reach a pseudo steady state. Do the authors have some deeper cores to show that carbon density in the top 30 cm is representative of long-term sequestration? Along these same lines – some of the plant communities looked at in this study (e.g. mangroves) can put "new" carbon into depths below 30cm through root production. The combination of these factors may lead to erroneous definition of carbon stocks as "blue carbon".

2. I wonder how applicable the use of contemporary variables is to the assessment of carbon stocks that are assumedly a function of conditions over the last several decades. This might be worth considering, and could explain the 50% of variability unaccounted for by the model outcomes. For example, community composition is changing in temperate regions such as the study area in this paper. Assuming a sediment accretion rate similar to SLR ($\sim$ 3mm/yr) – the 30cm soil profile used in the model integrates $\sim$ 100 years of environmental, ecological and anthropogenic conditions. Could this discrepancy in the temporal scale used for the predictor variables and carbon stock accumulation be an issue? 3. It would be good to see some kind of power analysis to assess whether the sample size is appropriate. I note there are R packages to do this for this kind of modelling approach.

Minor comments: Title – see comment 1 above, I am not sure the paper really assesses blue carbon due to the shallow sediment profile analysed. Also as this is a regional study, I think it might be appropriate to include something to clarify that in the title.

Abstract – Aims 1 and 2 should include the term regional, as the paper doesn't really produce a model that is applicable beyond the region of the study area

Abstract – last sentence. Without testing the validity of the modelling method to other regions, I am not sure this statement can be made. Suggest removing this statement or validating the modelling method elsewhere

Introduction Ln 40-43 Stocks of carbon are not directly related to greenhouse gas inventories which are based on flux rates.

Line 86 – Best not to start a new paragraph with "However" as this is a conjunction for connecting sentences within a paragraph.

Materials and methods Ln 153-159 Can the authors expand upon the methods used, including accuracy of analytical methods, the number of samples analysed by FT-MIR vs EA, and the results of cross validation between these 2 methods.

Ln 168 – 172 Assigning catchment characteristics to estuarine communities makes sense, but looking at Figure 1 most samples were collected from coastal embayment's. How might the influence from multiple catchments affect the model?

Ln 182-187 What was the vertical accuracy of the DEM? Considering the small elevation gradients across the intertidal zone, this is important.

Ln 210 – Is the 2001 data the most recent, or is this a typo?

Ln 215 Why was 30cm chosen as the depth representative of blue carbon stocks (see also earlier comment)? Can the authors add a few comments about this?

Results Throughout the results and the discussion the error associated with all estimates should be included. This error should combine model error and uncertainty in the spatial coverage of habitat areas.

Discussion Ln 434 As with the use of "however" to start a paragraph, "Further" should also be avoided.

452- 454 See earlier comment regarding applicability of this modelling framework to global assessments.

Data availability – I would like to see all of the underlying data made publically available rather than just the model outputs. These data can easily be attached as a supplementary.

---

## Referee Comment (RC2) · Anonymous Referee #2 · 25 Oct 2019

The manuscript 'Drivers and modelling of blue carbon stock variability' reports on the main drivers of sedimentary OC stocks (ecosystem – geomorphological – anthropogenic) in the top 30 cm of blue C ecosystems (tidal marshes, mangroves and sea grass meadows) across the state of Victoria (Australia). In addition, the authors used different general linear mixed-effects models to predict the spatial distribution of topsoil OC stocks of tidal wetlands in this region. The authors used a dataset they previously constructed (Ewers Lewis et al. 2018) of 287 sediment cores down to 30 cm depth to perform their analyses. The main results of the study show that ecological drivers (i.e. ecosystem type and dominant vegetation species) best explain the variability in C stocks, better than geomorphological and anthropogenic drivers. In addition, the authors calculate the regional topsoil C stock in tidal wetlands in Victoria to be 2.31

million ton C while identifying 'regions of interest', storing a substantial portion of total C in the studied region.

The manuscript is well-written and reads smoothly. The introduction provides a good overview of different factors controlling sedimentary OC stocks in vegetated coastal ecosystems that have been identified in literature. The material and methods section gives a clear overview of the study site, the data used and how the different models have been constructed, aided by a figure visualizing the workflow. The results section describes the most important results in a concise way and the discussion section frames the results with respect to existing literature. Overall, this manuscript provides an interesting approach to calculating sedimentary C stocks in blue C ecosystems at a large spatial scale and is well-worth publishing.

General comments

My main concern with the current manuscript is that it addresses topsoil C stocks, while it reports on 'blue C stocks' throughout the manuscript, without referring to the topsoil aspect. Emphasizing this aspect is, however, important: it is well known that sampling depth can have a large effect on conclusions drawn on relative differences in C stocks between coastal sediments at different locations or in different ecosystems. For example, depending on ecosystem-specific conditions, C stocks are known to decrease substantially with depth below the surface in certain ecosystems, while in others C stocks remain relatively constant with depth. Therefore, I would invite the authors to stress this aspect more throughout the manuscript: (i) the title would be more informative by including that the study concerns topsoil C stocks and (ii) the discussion should include a section where the implications of only considering topsoil samples is discussed.

Although I greatly appreciate that the authors have provided a statement that data is available upon request, I would like to ask the authors to consider publishing the data together with the manuscript, or making it available through an online repository, so

references to the data can be made. Open data is becoming increasingly important and has the potential to greatly advance the field of wetland biogeochemistry.

Specific comments

L83: allochthonous C can also come from terrestrial or estuarine sources

L149-150: would be good to provide a justification of why only the top 30 cm has been sampled

L154-155: would be good to report on the uncertainty associated with the use of spectroscopic techniques to estimate the 'C contents' of the samples. Any idea about the magnitude of this uncertainty? How were C stocks calculated? Were depth profiles of bulk density collected as well? Please briefly explain this, as this is important for the interpretation of the uncertainty on your results.

L237: I would refer to table S4 here; this will help the reader to understand how the models were constructed

L244: it's not clear from the text how the 'averaged models' were obtained and what these exactly are, please explain this in more detail

L298: it is not clear what you mean with 'intercept'

L333: Would be good to provide a measure of uncertainty on the total calculated C stock, similar to the standard deviations you report on the calculated numbers further down in this paragraph.

L336: Please briefly explain how the standard deviations were calculated. What do they exactly represent? Only the spatial variation within these ecosystems, or also uncertainties related to the model procedures used?

L411: I would suggest changing this title to 'Modelled topsoil blue C stocks'

Tables and figures

**BGD**

Table 2 and 3: it would be good to refer to where the description of the global models can be found (Table S4) in the caption (after '(global model 11, 5, 2 and 8)')

Table 4: I would change the caption to: '[. . .] and calculated C stocks [. . .]'

Table 5: I would change the caption to: 'Calculated blue C stocks [. . .]'

Figure 4: Please explain in the caption what the error bars represent (standard deviation?) and how they should be interpreted.

Technical corrections

L40: remove 'our'

L250: variable => variables

---

## Author Comment (AC1) · 28 Dec 2019

We thank Anonymous Referee #1 for their thoughtful comments (see Interactive Comment published on 01 Oct 2019 by Anonymous Referee #1). Below, we have responded to each comment and included changes to the manuscript text based on the feedback from Anonymous Referee #1.

**In this paper, the authors look to create a framework for modelling shallow carbon stocks (0-30cm) in vegetated coastal ecosystems. They use a combination of geomorphological, anthropogenic and ecological variables, combined with carbon stocks from a large number of shallow cores (n = 287) to construct the model and estimate carbon stocks for a region in southern Australia. The model could account for ~ 49% of the variability in shallow carbon stocks, with plant community being the strongest predictor in the model.**

**Generally this is an interesting paper, but I have some points that should be clarified.**

**1. Can these shallow carbon stocks be considered "blue carbon"? There is growing evidence that carbon within the surface layers is still highly susceptible to degradation. A typical profile of carbon down the soil profile in these vegetated habitats show a decline with depth until reach a pseudo steady state. Do the authors have some deeper cores to show that carbon density in the top 30 cm is representative of long-term sequestration? Along these same lines – some of the plant communities looked at in this study (e.g. mangroves) can put "new" carbon into depths below 30cm through root production. The combination of these factors may lead to erroneous definition of carbon stocks as "blue carbon".**

"Blue Carbon" is broadly defined in the scientific literature as organic carbon captured and stored in ocean ecosystems, especially mangrove forests, seagrass meadows, and salt marshes (Macreadie et al., 2019; Mcleod et al., 2011; Nellemann et al., 2009), and includes carbon pools in the living plant biomass (above- and belowground), dead plant biomass (e.g. leaf litter, dead wood), and sediments (Mcleod et al., 2011). Though carbon stocks associated with sediments are often considered more "permanent" relative to living biomass, both biomass and sediments (including shallow portions of sediments) are defined as "blue carbon".

Across depths, sediments may span a continuum of carbon quality from "labile" to "recalcitrant" or "refractory" based on a number of criteria, which involve more than depth alone (Lovelock et al.). By these criteria, virtually all carbon is susceptible to degradation to some degree. However, this does not negate the fact that carbon present in surface sediments represents the offsetting capacity of blue carbon sediments (potential long-term carbon storage), and is an important measure of carbon that may be more vulnerable to remineralization following disturbance.

To be very clear that we are not referring to the entire sediment C pool, we have altered our wording throughout the entire paper to specify "shallow sediment C stocks" in place of "C stocks" or "sediment C stocks". This includes the title, which has been changed from *Drivers and modeling of blue carbon stock variability* to *Drivers and modeling of blue carbon stock variability in shallow sediments of southeast Australia*, as well as section headers and within the entirety of the text.

We have added text to the methods to explain our rationale for the study design, including the use of 30 cm cores:

*Though it is common in the literature to sample to 1 m deep in blue C sediments, the sampling protocol used for collecting these data (Ewers Lewis et al., 2018) was designed to maximize spatial coverage of shallow sediment C samples rather than sample entire sediment profiles (which may extend well beyond 1 meter deep). Greater spatial coverage allowed us to test the relationships between a variety of potential drivers and surface sediment C stocks on both fine and broad scales.*

We have also added text to the discussion to explain why the top 30 cm are the most relevant sediments for assessing the impact of contemporary environmental factors on shallow sediment C stocks (please see new discussion text under our response to your second comment, which directly addresses this point).

Thank you for bringing up the points about "new" C at depth and the relationship between shallow and deep C stocks. We recognize that surface stocks may not always represent stocks at depth, and also that processes happening at the surface can impact deeper sediments, and have added text about this topic to the discussion:

*Modern-day factors influencing vegetation can also have impacts on C stocks deeper than the sediments we measured. The effects of underground biomass on sediment C stocks can extend beyond the top 30 cm, and in fact new C inputs and active C cycling by microbial communities can occur as deep as underground roots extend (Trumbore, 2009). These new C additions (and fluxes) at depth fall outside the general pattern of sediment C decay down-core in vegetated ecosystems (Trumbore, 2009) which has previously allowed for linear or logarithmic regressions to be used to extrapolate 1-m deep C contents from shallow (e.g. 30-50 cm deep) sediment C data (Macreadie et al., 2017a; Serrano et al., 2019). The activity of underground biomass and microbes at depth, when considered over space and time, may account for large C fluxes. The influence of anthropogenic activities, such as land use changes, on these processes via impacts to vegetation may largely go unnoticed based on current methods (Trumbore, 2009), both in this study and in blue C stock assessments on larger scales. We suggest further research to understand the dynamics of active C cycling at sediment depths traditionally considered stable.*

Later in the discussion we have also added:

*It is important to emphasize here that total sediment depths in blue C ecosystems can vary greatly, and are commonly deeper than 30 cm. Blue C ecosystems can have sediments up to several meters deep (e.g. Lavery et al., 2013; Scott and Greenberg, 1983), suggesting the estimates of C stocks measured here are conservative. In spite of these limitations, surface sediment C stock estimates give us valuable knowledge about the sediment C pool most vulnerable to disturbance and how it may be impacted by environmental drivers.*

**2. I wonder how applicable the use of contemporary variables is to the assessment of carbon stocks that are assumedly a function of conditions over the last several decades. This might be worth considering, and could explain the 50% of variability unaccounted for by the model outcomes. For example, community composition is changing in temperate regions such as the study area in this paper. Assuming a sediment accretion rate similar to SLR (~ 3mm/yr) – the 30cm soil profile used in the model integrates ~ 100 years of environmental, ecological and anthropogenic conditions. Could this discrepancy in the temporal scale used for the predictor variables and carbon stock accumulation be an issue?**

Thank you for this comment. We have added paragraphs to the discussion to describe the challenges associated with matching temporal time points in the sediments to contemporary variables and how the depth we chose and regional accretion rates might shine light on some of these issues:

*We also aimed to maximize our ability to capture relationships between contemporary drivers and sediment C stocks by utilizing sediment C stock data to only 30 cm deep, a sediment horizon more directly impacted by recent environmental conditions compared to deeper stocks due to age. Based on previously estimated sediment accretion rates in blue C ecosystems in the study region (averaging 2.51 to 2.66 mm year$^{-1}$ in tidal marshes (Ewers Lewis et al., 2019; Rogers et al., 2006a) and 7.14 mm year$^{-1}$ in mangroves (Rogers et al., 2006a)), the top 30 cm of sediment represents roughly ~113-120 years of accretion in Victorian tidal marshes and ~42 years of accretion in Victorian mangroves. These time scales suggest sediments depths utilized in this study are more appropriate for assessing the impacts of modern environmental conditions on sediment C stocks compared to meter-deep stocks, which can be thousands of years old (e.g. Ewers Lewis et al., 2019). Using shallow sediment C stocks also allows us to be more confident that the vegetation present now has been there during the time of sediment accretion, unlike deeper sediments that are thousands of years old and for which it is difficult to determine what vegetation, if any, was present at the time of accretion.*

*The variability in shallow sediment C stocks that could not be explained by our modeling may also be related to the inherent challenges surrounding spatial and temporal matching of driver proxies and sediment C stock measurements; the relationship between shallow sediment C stocks and contemporary environmental settings can be represented more accurately for some variables over others.*

*Ecosystem type was a relatively powerful predictor of shallow sediment C stock variability in our study and this is likely due, in part, to the direct relationship between vegetation and surface sediments. In most vegetated ecosystems, the majority of underground plant biomass and microbial activity exists within the top 20 cm of soils (Trumbore, 2009). For saltmarsh, it has been demonstrated that the top 30 cm of sediment are directly impacted by current vegetation (Owers et al., 2016). Therefore, using shallow sediment C stock measurements allowed us to take advantage of the direct relationship between vegetation and C stocks to explain variability in surface sediments.*

*The portion of recently accreted sediments influenced by contemporary anthropogenic drivers is harder to identify than that of ecosystems. Based on estimated accretion rates for this region from the literature (Ewers Lewis et al., 2019; Rogers et al., 2006b), 30 cm deep sediments would have taken an average of ~80 years to accumulate in Victoria (~117 years in tidal marsh and ~42 years in mangroves). Though sedimentation rates vary over time, they are relatively steady in comparison to changes in anthropogenic drivers, such as land use change. This means that modern day maps of land use, though useful for looking at the general impact of various activities, may be more useful for relating to variability in sediment C stocks when the data is assessed at a finer resolution. For example, comparing land use area data across various time periods with C densities in aged bands of sediment could help capture the pulse effects of sudden land use changes in narrower sediment horizons representative of the same time periods. In this study, the effects of land-use change may have been too diluted within the 30-cm horizons to relate to impacts on sediment C stock.*

**3. It would be good to see some kind of power analysis to assess whether the sample size is appropriate. I note there are R packages to do this for this kind of modelling approach.**

Using a simpler modelling approach, a power analysis was conducted on this dataset for a separate study using the SIMR package. This analysis showed that a power of 80% was reached across all ecosystems within the sample sizes of this dataset (Young, M. A., Macreadie, P. I., Duncan, C., Carnell, P. E., Nicholson, E., Serrano, O., Duarte, C. M., Shiell, G., Baldock, J. and Ierodiaconou, D. 2018. Optimal soil carbon sampling designs to achieve cost-effectiveness: a case study in blue carbon ecosystems, Biol. Lett., 14(20180416), doi:10.1098/rsbl.2018.0416).

Though we like the idea of running a power analysis for this more complex modelling approach, we have not been able to find an R package that is specifically compatible with the information theory multimodel approach we have taken. Though there are R packages for doing power analysis on glmm/glmer models created in the R package lme4 (e.g. SIMR), we have not found a package compatible with averaged models that have been generated by dredging and averaging general linear mixed effects models (made with the R package MuMIn).

To clarify why we used the more complex modelling approach (AICc with model averaging), we added the following paragraph to the methods, which explains the better accuracy of predictions generated from averaged models compared to traditional approaches utilizing a single "best" model (e.g. glmer in the lme4 package):

*To identify drivers of shallow sediment C stock variability and create the best predictive model of shallow sediment C stocks to 30 cm deep we utilized a multi-step process based on an information theoretic approach and multimodel inference (Figure 3). Traditional approaches have relied on identification of the "best" data-based model; however, information-theoretic approaches allow for more reliable predictions through utilization of multiple models, especially in cases where lower ranked models may be essentially as good as the best (Burnham and Anderson, 2002; Symonds and Moussalli, 2011). Further, information theoretic model selection has been demonstrated to provide significant advantages for explaining phenomena with more complex drivers (Richards et al., 2011). Here, we first looked broadly at our variables of interest by narrowing down to the best models containing all possible variables ("global" models, as explained below) using AICc (Akaike information criteria, corrected for small sample size) to explain the variability observed in the training dataset (70% of total C stock data; Symonds and Moussalli, 2011). From there, we identified which variables within the best global models best explained the observed variability in C stock data in order to remove unnecessary variables from the model equation (through the process of "dredging" and selecting the best subset, explained in detail below). The validity of removing unnecessary variables from the model is supported by the concept of parsimony, which suggests models more complicated than the best model provide little benefit and should be eliminated (Burnham and Anderson, 2002; Richards, 2008). The best subset of models generated from the global models ("dredge products") were selected based on delta AICc<2, which are viewed as essentially interchangeable with the best model (Symonds and Moussalli, 2011). Each subset of best models was used to generate an averaged model, which was tested by generating predictions of C stocks for a reserved (30%) subset of the dataset. The best performing model was used to generate a predictive map of C stocks to 30 cm deep for mapped blue C ecosystems in Victoria.*

**Minor comments: Title – see comment 1 above, I am not sure the paper really assesses blue carbon due to the shallow sediment profile analysed. Also as this is a regional study, I think it might be appropriate to include something to clarify that in the title.**
Thank you for this feedback. We have changed the title to better reflect the specifics of our study by changing it from "*Drivers and modelling of blue carbon stock variability*" to "*Drivers and modelling of blue carbon stock variability in shallow sediments of southeast Australia*".

**Abstract – Aims 1 and 2 should include the term regional, as the paper doesn't really produce a model that is applicable beyond the region of the study area**

We have updated the aims as suggested by adding the phrase "… in southeast Australia" to each of them (in both the abstract and introduction).

**Abstract – last sentence. Without testing the validity of the modelling method to other regions, I am not sure this statement can be made. Suggest removing this statement or validating the modelling method elsewhere.**

Thank you for this comment. We have modified this sentence to reflect the need to test the validity of using the modelling methods elsewhere.

The sentence previously read:
*Globally these methods can be applied to identify relationships between environmental drivers and C stocks to produce predictive C stock models at scales relevant for resource management.*

And has been changed to:
*We recommend these methods be tested for applicability to other regions of the globe for identifying drivers of C stock variability and producing predictive C stock models at scales relevant for resource management.*

**Introduction Ln 40-43 Stocks of carbon are not directly related to greenhouse gas inventories which are based on flux rates.**

We are aware of this, but also recognize that knowledge of stocks enables for more accurate estimates of fluxes, which are related to stocks (e.g. the maximum amount of carbon that can be remineralized and converted to CO2 if subjected to disturbance is the total stock present). We have added "and fluxes" to this sentence to ensure we convey the importance of fluxes, which relate C stocks and greenhouse gas inventories to one another:
*With the current momentum for including blue C ecosystems in global greenhouse gas inventories, there is a need to quantify the magnitude of C stocks and fluxes…*

**Line 86 – Best not to start a new paragraph with "However" as this is a conjunction for connecting sentences within a paragraph.**

This has been removed.

**Materials and methods Ln 153-159 Can the authors expand upon the methods used, including accuracy of analytical methods, the number of samples analysed by FT-MIR vs EA, and the results of cross validation between these 2 methods.**

Thank you for this request. We have added more details, such as those requested, to make our methods more transparent and easier to understand without referencing the original C stock paper, as follows:

*Sediment C stocks to 30 cm deep (referred to throughout the paper as "shallow sediment C stocks") were estimated for 287 sediment cores from 96 blue C ecosystems across Victoria in southeast Australia (Ewers Lewis et al., 2018; Figure 1). Full details of sample collection, laboratory analyses, and calculations of C stocks can be found in Ewers Lewis et al. (2018). Briefly, three replicate sediment cores (5-cm inner diameter) were taken in each ecosystem (n=125 in tidal marsh, n=60 in mangroves, and n=102 in seagrasses). Once back in the laboratory, samples were taken from three depths (0-2, 14-16, 28-30 cm) within each core. Samples were dried at 60℃ until a consistent weight was achieved, then ground. Dry bulk density (DBD) was calculated as the dry weight divided by the original volume for all samples.*

*Based on the protocols by Baldock et al. (2013), a combination of diffuse reflectance Fourier transform mid-infrared (MIR) spectroscopy and elemental analysis via oxidative combustion using a LECO Trumac CN analyzer was used to determine organic C contents of all samples. Previous studies have demonstrated the accuracy of using MIR to estimate organic C stocks of sediments (Baldock et al., 2013; Van De Broek and Govers, 2019; Ewers Lewis et al., 2018). MIR spectra were acquired for all samples, then a subset of 200 representative samples was selected based on a principle components analysis (PCA) of the MIR results utilizing the Kennard-Stone algorithm. Gravimetric contents of organic carbon were measured directly in the laboratory for the 200-sample subset (Baldock et al. 2013). A partial least squares regression (PSLR) was created using a Random Cross Validation Approach (Unscrambler 10.3, CAMO Software AS, Oslo, Norway) and used to build algorithms to predict square root transformed total carbon, total organic carbon, total nitrogen, and inorganic carbon for the entire dataset. The PSLR model was evaluated based on parameters from the chemometric analysis of soil properties (Bellon-Maurel et al., 2010; Bellon-Maurel and McBratney, 2011), and the relationship between measured and predicted values was assessed based on slope, offset, correlation coefficient (r), R-squared, the root mean square error (RMSE), bias, and the standard error (SE) of calibration (SEC) and validation (SEP; see Ewers Lewis et al., 2018 for full details). R-squared values for all square root transformed variables were ≥0.94.*

*Sediment C stocks were calculated based on Howard et al., 2014. Organic C density (mg C cm$^{-3}$) was calculated by multiplying organic C content (mg C g$^{-1}$) by DBD (g cm$^{-3}$). Linear splines were applied to each core to estimate C density for each 2 cm increment within the 30 cm core, then C densities for each interval (measured and extrapolated) were summed and converted to Mg C ha$^{-1}$ to estimate total stock down to 30 cm deep for each core location…*

**Ln 168 – 172 Assigning catchment characteristics to estuarine communities makes sense, but looking at Figure 1 most samples were collected from coastal embayment's. How might the influence from multiple catchments affect the model?**
The averaged model that we used to make the C stock prediction maps included ecosystem type, slope, and distance to coast, none of which we would expect to be substantially affected by the influence of multiple catchments. The influence of anthropogenic activities on surface sediment C stocks in locations receiving inputs from multiple locations, on the other hand, would be very difficult to track, and could have affected our ability to identify a relationship between anthropogenic drivers and C stock variability in our models. We have added the following text to the discussion to address this point, as follows:

*Spatially, anthropogenic variables are also difficult to assign to particular ecosystem locations or depths. Many blue C ecosystems in Victoria are located on coastal embayments and receive inputs from multiple catchments, making the influence of specific areas of land-use or population changes difficult to track to specific ecosystem locations.*

**Ln 182-187 What was the vertical accuracy of the DEM? Considering the small elevation gradients across the intertidal zone, this is important.**
Thank you for this question. We have added the following text to the methods section description of the elevation data, as follows, to clarify the accuracy of the DEM:

*The elevation data are a composite product that integrated terrestrial and bathymetric LIDAR as well as multibeam sonar data. The vertical accuracies of the data varied with sensor setup for acquisition: ±10 cm at 1 sigma (68% conf. level) in bare ground for topographic LIDAR data (for the majority of our dataset), ±50 cm for bathymetric LIDAR, and ±<10 cm for multibeam sonar data.*

**Ln 210 – Is the 2001 data the most recent, or is this a typo?**
This should say 2011 – thank you for bringing this to our attention.

**Ln 215 Why was 30cm chosen as the depth representative of blue carbon stocks (see also earlier comment)? Can the authors add a few comments about this?**
Thank you for this question. We have added a paragraph in the methods to address this topic and have also added a paragraph about the implications of measuring to 30 cm in our discussion section.

Added to methods:
*Though it is common in the literature to sample to 1 m deep in blue C sediments, the sampling protocol used for collecting these data (Ewers Lewis et al., 2018) was designed to maximize spatial coverage of shallow sediment C samples rather than sample entire sediment profiles (which may extend well beyond 1 meter deep). Greater spatial coverage allowed us to test the relationships between a variety of potential drivers and surface sediment C stocks on both fine and broad scales.*

Added to discussion:
*We also aimed to maximize our ability to capture relationships between contemporary drivers and sediment C stocks by utilizing sediment C stock data to only 30 cm deep, a sediment horizon more directly impacted by recent environmental conditions compared to deeper stocks due to age. Based on previously estimated sediment accretion rates in blue C ecosystems in the study region (averaging 2.51 to 2.66 mm year$^{-1}$ in tidal marshes (Ewers Lewis et al., 2019; Rogers et al., 2006a) and 7.14 mm year$^{-1}$ in mangroves (Rogers et al., 2006a)), the top 30 cm of sediment represents roughly ~113-120 years of accretion in Victorian tidal marshes and ~42 years of accretion in Victorian mangroves. These time scales suggest sediments depths utilized in this study are more appropriate for assessing the impacts of modern environmental conditions on sediment C stocks compared to meter-deep stocks, which can be thousands of years old (e.g. Ewers Lewis et al., 2019). Using shallow sediment C stocks also allows us to be more confident that the vegetation present now has been there during the time of sediment accretion, unlike deeper sediments that are thousands of years old and for which it is difficult to determine what vegetation, if any, was present at the time of accretion.*

**Results Throughout the results and the discussion the error associated with all estimates should be included. This error should combine model error and uncertainty in the spatial coverage of habitat areas.**
Model error is presented through model validation in our study. We reserved 30% of our C stock dataset to test the accuracy of the predictions generated by each averaged model (as described in sections 3.2 of the results and displayed in Figure S3). These results showed that our best model explained ~half the observed variability in C stocks (Adj R-sq=0.4881; averaged model 2). We have clarified the errors associated with our model predictions by providing further details on the outputs of our validation step, which utilized the reserved dataset (30% of our original carbon dataset) for assessing the prediction power of each averaged model. Using linear regression, we compared predicted values from each averaged model to actual measured C stock values for reserved dataset. The complete outputs for this analysis have been added to the results section 3.2 ("Model validation"), as follows:

*Linear regressions of predicted versus actual measured shallow sediment C values produced the following outputs for each averaged model: averaged model 11, residual standard error (RSE)=38.36 on 84 degrees of freedom (df), adjusted R-squared (R-sq(adj))=0.4868, F-statistic(F-stat)=81.63 on 1 and 84 df, p-value=5.044e-14; averaged model 5, RSE=38.51, R-sq(adj)=0.4829, F-stat=80.39 on 1 and 84 df, p-value=6.953e-14; averaged model 2, RSE=38.32, R-sq(adj)=0.4881, F-stat=82.06 on 1 and 84 df,*

*p-value=4.517e-14; averaged model 10, RSE=39.67, R-sq(adj)=0.4514, F-stat=70.93 on 1 and 84 df, p-value=8.645e-13; averaged model 4, RSE=39.84, R-sq(adj)=0.4465, F-stat=69.58 on 1 and 84 df, p-value=1.254e-12; averaged model 1; RSE=39.48, R-sq(adj)=0.4566, F-stat=72.43 on 1 and 84 df, p-value=5.73e-13; averaged model 7, RSE=39.29, R-sq(adj)=0.4618, F-stat=73.94 on 1 and 84 df, p-value=3.81e-13.*

We have also been more explicit about the error associated with the averaged model used to generate our state-wide shallow sediment C stock predictions by adding text to the results section on modelled shallow sediment blue C stocks (section 3.3) as follows:

*We estimated a total of over 2.31 million Mg C stored in the top 30 cm of sediments in the ~68,700 ha of blue C ecosystems across Victoria (Table 4; Figure 5). This estimate is based on predictions from our best averaged model that utilized ecosystem type as the ecological variable (averaged model 7), which explained 46.18% of observed variability in C stock data and had an RSE of 39.29.*

Due to the complexity of our multimodel approach, we did not include standard errors in predicted values for the entire region of Victoria due to the combination of using an averaged model and having random effects (there are no compatible R packages, to our knowledge).

For spatial coverage, there is no data available to estimate uncertainty in habitat area that gives the specificity we would need to alter our predictions (i.e. we would have to know the exact locations due to the spatially explicit nature of the model). We used the most recent and complete maps available for each ecosystem for Victoria, which did not include specific locations where coverage was questionable. The main uncertainty stems from the potential changes in habitat area since the time of mapping or errors in mapping, and we cannot measure that error because no more-recent maps are available.

**Discussion Ln 434 As with the use of "however" to start a paragraph, "Further" should also be avoided.**
We thank you for the suggestion. We have moved this sentence up to be included in the previous paragraph, and moved the remainder of the paragraph to an earlier section of the text to improve flow.

**452- 454 See earlier comment regarding applicability of this modelling framework to global assessments.**
We understand your point and have edited the text to reflect that this framework needs testing in other regions for applicability.

The text previously read:
*Globally, these methods are applicable for identifying relationships between potential environmental drivers and C stocks for creating predictive C stock models in blue C ecosystems at scales relevant for resource management applications.*

And has been changed to:
*We recommend these methods be tested in other areas of the globe to determine whether they may be applicable for identifying relationships between potential environmental drivers and C stocks for creating predictive C stock models in blue C ecosystems at scales relevant for resource management applications in other regions.*

**Data availability – I would like to see all of the underlying data made publically available rather than just the model outputs. These data can easily be attached as a supplementary.**

To improve transparency and accessibility of the data both utilized and produced in this study we have added the following table to the supplements (below) with a reference to it in the methods section:

*Complete details of data availability for inputs and outputs of our models can be found in supplementary Table S10.*

Any modifications made to these data for producing our models are described in the methods section of this manuscript.

Please also note the data produced in this study (R code and model prediction rasters) will be uploaded to an online repository upon acceptance of this manuscript for publication, due to both the large size of the raster files (making them too large for supplementary information) and the intellectual property associated with this work as part of the first author's Ph.D. dissertation. We have inquired about joining and depositing our data to the Coastal Carbon Research Coordination Network (https://serc.si.edu/coastalcarbon/join-the-network) Data Clearinghouse so that the data can be archived in a Smithsonian Library digital repository and accessible through a digital object identifier (DOI).

**Table S10.** Data availability

| Data Item | Description | Data Source & Location |
|---|---|---|
| Carbon Stock Dataset | Percent organic carbon and dry bulk density data for sediment sampled to 30 cm deep in 96 blue carbon ecosystems (saltmarshes, mangrove forests, and seagrass meadows) across Victoria, Australia. | Ewers Lewis et al. 2018; Deakin Research Online Deakin University's Data Repository https://dro.deakin.edu.au/view/DU:30093405 |
| Ecosystem Extent Vectors | 1. Mangrove areal extent in Victoria, Australia; saltmarsh areal extent and ecological vegetation classes in Victoria, Australia.
2. Seagrass areal extent in the major bays and estuaries of Victoria, Australia.
    a. Port Phillip Bay
    b. Western Port Bay
    c. Corner Inlet and Nooramunga
    d. Gippsland Lakes
    e. Minor Inlets of Victoria | 1. Boon et al. 2001; OzCoasts Australian Online Coastal Information, Victorian Saltmarsh and Mangrove Vegetation Maps https://ozcoasts.org.au/geom_geol/vic/Saltmarsh/Master
2. Available from:
a. Ball et al., 2014; Blake and Ball, 2001a https://discover.data.vic.gov.au/dataset/port-phillip-bay-1-25-000-seagrass-2000
b. Blake and Ball, 2001b Distribution of Seagrass in Western Port in 1999 https://discover.data.vic.gov.au/dataset/distribution-of-seagrass-in-western-port-in-1999
c. Roob et al., 1998 Corner Inlet Seagrass 1998 https://discover.data.vic.gov.au/dataset/corner-inlet-seagrass-1998
d. Roob and Ball, 1997 Gippsland Lakes Seagrass 1997 https://discover.data.vic.gov.au/dataset/gippsland-lakes-seagrass-1997
e. Blake et al., 2000 Anderson Inlet Seagrass 1999 https://discover.data.vic.gov.au/dataset/anderson-inlet-seagrass-1999 Tamboon Inlet Seagrass 1999 |

| | | https://discover.data.vic.gov.au/dataset/tamboon-inlet-seagrass-1999
Wingan Inlet Seagrass 1999
https://discover.data.vic.gov.au/dataset/wingan-inlet-seagrass-1999
Shallow Inlet Seagrass 1999
https://discover.data.vic.gov.au/dataset/shallow-inlet-seagrass-1999
Mallacoota Inlet Seagrass 1999
https://discover.data.vic.gov.au/dataset/mallacoota-inlet-seagrass-1999
Sydenham Inlet Seagrass 1999
https://discover.data.vic.gov.au/dataset/sydenham-inlet-seagrass-1999 |
|---|---|---|
| Elevation Raster | A gap free digital elevation model (DEM) for the coastal region of Victoria, Australia, that combines 2.5 m and 10 m DEMs. | Victorian Coastal Digital Elevation Model (VCDEM 2017)
https://vmdp.deakin.edu.au/geonetwork/srv/eng/metadata.show?uuid=8d3ccf63-ee85-41cd-917e-933624a50b2e |
| Freshwater Vectors | Location of channels and other freshwater objects in Victoria, Australia. | Vicmap Hydro 1:25,000
Victorian Government Data portal
https://discover.data.vic.gov.au/dataset/vicmap-hydro-1-25-000 |
| Coastline Vector | Line delineating the coastline of Victoria, Australia. | Victorian Coastline 2008
Victorian Government Data portal
https://discover.data.vic.gov.au/dataset/victorian-coastline-2008 |
| Lithology Vectors | Rock types across Victoria, Australia. | Geomorphology of Victoria
Victorian Government Data portal
https://discover.data.vic.gov.au/dataset/geomorphology-of-victoria |
| Land Use Vectors | Primary land use designations for land parcels in Victoria, Australia. | Victorian Land Use Information System 2014/2015
Victorian Government Data portal
https://discover.data.vic.gov.au/dataset/victorian-land-use-information-system-2014-2015 |
| Population Raster | Human population data for Victoria, Australia. | Australian Population Grid, 2011
Australian Bureau of Statistics
https://www.abs.gov.au/AUSSTATS/abs@.nsf/Lookup/1270.0.55.007Main+Features12011?OpenDocument |
| R Code | R code used to identify drivers and model carbon shallow sediment carbon stocks. | This study.
Data depository TBD upon acceptance for publication. |
| Model Output Raster | Shallow sediment (to 30 cm deep) carbon stock predictions in blue carbon ecosystems (seagrass meadows, mangrove forests, and saltmarshes) in Victoria, Australia | This study.
Data depository TBD upon acceptance for publication. |

---

## Author Comment (AC2) · 28 Dec 2019

We thank anonymous referee #2 for their kind words and helpful insights on the present manuscript (see Interactive Comment published on 25 October 2019 by Anonymous Referee #2). Below, we respond to each comment individually and describe how this feedback has been used to improve the manuscript.

**The manuscript 'Drivers and modelling of blue carbon stock variability' reports on the main drivers of sedimentary OC stocks (ecosystem – geomorphological – anthropogenic) in the top 30 cm of blue C ecosystems (tidal marshes, mangroves and sea grass meadows) across the state of Victoria (Australia). In addition, the authors used different general linear mixed-effects models to predict the spatial distribution of topsoil OC stocks of tidal wetlands in this region. The authors used a dataset they previously constructed (Ewers Lewis et al. 2018) of 287 sediment cores down to 30 cm depth to perform their analyses. The main results of the study show that ecological drivers (i.e. ecosystem type and dominant vegetation species) best explain the variability in C stocks, better than geomorphological and anthropogenic drivers. In addition, the authors calculate the regional topsoil C stock in tidal wetlands in Victoria to be 2.31 million ton C while identifying 'regions of interest', storing a substantial portion of total C in the studied region.**

**The manuscript is well-written and reads smoothly. The introduction provides a good overview of different factors controlling sedimentary OC stocks in vegetated coastal ecosystems that have been identified in literature. The material and methods section gives a clear overview of the study site, the data used and how the different models have been constructed, aided by a figure visualizing the workflow. The results section describes the most important results in a concise way and the discussion section frames the results with respect to existing literature. Overall, this manuscript provides an interesting approach to calculating sedimentary C stocks in blue C ecosystems at a large spatial scale and is well-worth publishing.**

Thank you very much.

**General comments**

**My main concern with the current manuscript is that it addresses topsoil C stocks, while it reports on 'blue C stocks' throughout the manuscript, without referring to the topsoil aspect. Emphasizing this aspect is, however, important: it is well known that sampling depth can have a large effect on conclusions drawn on relative differences in C stocks between coastal sediments at different locations or in different ecosystems. For example, depending on ecosystem-specific conditions, C stocks are known to decrease substantially with depth below the surface in certain ecosystems, while in others C stocks remain relatively constant with depth. Therefore, I would invite the authors to stress this aspect more throughout the manuscript: (i) the title would be more informative by including that the study concerns topsoil C stocks and (ii) the discussion should include a section where the implications of only considering topsoil samples is discussed.**

We appreciate this point and agree that it is important we make it more clear that we are referring to the top 30 cm of sediments. We have done this throughout the manuscript. First, the title has been changed from *Drivers and modeling of blue carbon stock variability* to *Drivers and modeling of blue carbon stock*

variability **in shallow sediments of southeast Australia**. Next, we have altered our wording throughout the entire text and in the section headings to specify "shallow sediment C stocks" in place of "C stocks" or "sediment C stocks". We have clarified this point in the methods also:

[revised manuscript text omitted]

**Although I greatly appreciate that the authors have provided a statement that data is available upon request, I would like to ask the authors to consider publishing the data together with the manuscript, or making it available through an online repository, so references to the data can be made. Open data is becoming increasingly important and has the potential to greatly advance the field of wetland biogeochemistry.**

Thank you for this suggestion. We have inquired about joining and submitting data to the Coastal Carbon Research Coordination Network Data Clearinghouse (https://serc.si.edu/coastalcarbon), which can host our data in one of the Smithsonian Library's digital repositories and be included in the Coastal Carbon Atlas (https://ccrcn.shinyapps.io/CoastalCarbonAtlas/). The data will be issued a digital object identifier (DOI) that can be included with the published paper for readers to easily access the data.

Additionally, to improve transparency and accessibility of the data both utilized and produced in this study, we have added the following table to the supplements (below) with a reference to it in the methods section:

*Complete details of data availability for inputs and outputs of our models can be found in supplementary Table S10.*

Any modifications made to these data for producing our models are described in the methods section of this manuscript. Please note the reference to the data produced in this study (R code and model prediction rasters) will be updated in this table. Due to both the large size of the raster files and the intellectual property associated with this work as part of the first author's Ph.D. dissertation, the data will be hosted in an online repository (rather than as a supplement) at the time of publication.

**Table S10.** Data availability

| Data Item | Description | Data Source & Location |
|---|---|---|
| Carbon Stock Dataset | Percent organic carbon and dry bulk density data for sediment sampled to 30 cm deep in 96 blue carbon ecosystems (saltmarshes, mangrove forests, and seagrass meadows) across Victoria, Australia. | Ewers Lewis et al. 2018; Deakin Research Online Deakin University's Data Repository https://dro.deakin.edu.au/view/DU:30093405 |

| Ecosystem Extent Vectors | 1. Mangrove areal extent in Victoria, Australia; saltmarsh areal extent and ecological vegetation classes in Victoria, Australia.
2. Seagrass areal extent in the major bays and estuaries of Victoria, Australia.
  a. Port Phillip Bay
  b. Western Port Bay
  c. Corner Inlet and Nooramunga
  d. Gippsland Lakes
  e. Minor Inlets of Victoria | 1. Boon et al. 2001; OzCoasts Australian Online Coastal Information, Victorian Saltmarsh and Mangrove Vegetation Maps
https://ozcoasts.org.au/geom_geol/vic/Saltmarsh/Master
2. Available from:
a. Ball et al., 2014; Blake and Ball, 2001a
https://discover.data.vic.gov.au/dataset/port-phillip-bay-1-25-000-seagrass-2000
b. Blake and Ball, 2001b
Distribution of Seagrass in Western Port in 1999
https://discover.data.vic.gov.au/dataset/distribution-of-seagrass-in-western-port-in-1999
c. Roob et al., 1998
Corner Inlet Seagrass 1998
https://discover.data.vic.gov.au/dataset/corner-inlet-seagrass-1998
d. Roob and Ball, 1997
Gippsland Lakes Seagrass 1997
https://discover.data.vic.gov.au/dataset/gippsland-lakes-seagrass-1997
e. Blake et al., 2000
Anderson Inlet Seagrass 1999
https://discover.data.vic.gov.au/dataset/anderson-inlet-seagrass-1999
Tamboon Inlet Seagrass 1999
https://discover.data.vic.gov.au/dataset/tamboon-inlet-seagrass-1999
Wingan Inlet Seagrass 1999
https://discover.data.vic.gov.au/dataset/wingan-inlet-seagrass-1999
Shallow Inlet Seagrass 1999
https://discover.data.vic.gov.au/dataset/shallow-inlet-seagrass-1999
Mallacoota Inlet Seagrass 1999
https://discover.data.vic.gov.au/dataset/mallacoota-inlet-seagrass-1999
Sydenham Inlet Seagrass 1999
https://discover.data.vic.gov.au/dataset/sydenham-inlet-seagrass-1999 |
|---|---|---|
| Elevation Raster | A gap free digital elevation model (DEM) for the coastal region of Victoria, Australia, that combines 2.5 m and 10 m DEMs. | Victorian Coastal Digital Elevation Model (VCDEM 2017)
https://vmdp.deakin.edu.au/geonetwork/srv/eng/metadata.show?uuid=8d3ccf63-ee85-41cd-917e-933624a50b2e |
| Freshwater Vectors | Location of channels and other freshwater objects in Victoria, Australia. | Vicmap Hydro 1:25,000
Victorian Government Data portal
https://discover.data.vic.gov.au/dataset/vicmap-hydro-1-25-000 |
| Coastline Vector | Line delineating the coastline of Victoria, Australia. | Victorian Coastline 2008
Victorian Government Data portal
https://discover.data.vic.gov.au/dataset/victorian-coastline-2008 |
| Lithology Vectors | Rock types across Victoria, Australia. | Geomorphology of Victoria
Victorian Government Data portal |

| | | https://discover.data.vic.gov.au/dataset/geomorphology-of-victoria |
|---|---|---|
| Land Use Vectors | Primary land use designations for land parcels in Victoria, Australia. | Victorian Land Use Information System 2014/2015 Victorian Government Data portal https://discover.data.vic.gov.au/dataset/victorian-land-use-information-system-2014-2015 |
| Population Raster | Human population data for Victoria, Australia. | Australian Population Grid, 2011 Australian Bureau of Statistics https://www.abs.gov.au/AUSSTATS/abs@.nsf/Lookup/1270.0.55.007Main+Features12011?OpenDocument |
| R Code | R code used to identify drivers and model carbon shallow sediment carbon stocks. | This study. Data depository TBD upon acceptance for publication. |
| Model Output Raster | Shallow sediment (to 30 cm deep) carbon stock predictions in blue carbon ecosystems (seagrass meadows, mangrove forests, and saltmarshes) in Victoria, Australia | This study. Data depository TBD upon acceptance for publication. |

**Specific comments**

**L83: allochthonous C can also come from terrestrial or estuarine sources**

Thank you we have altered the wording to reflect that although we are referring to C transported into the ecosystem via marine tidal flooding, we recognize that C could have originally come from other sources (including, but not limited to terrestrial or estuarine sources), as follows:

*In higher elevations tidal flooding is less frequent, providing less opportunity for particles and C to settle out of the water column, resulting in a lower contribution of allochthonous C from marine or other sources compared to lower, more frequently inundated marshes…*

**L149-150: would be good to provide a justification of why only the top 30 cm has been sampled**

Thank you for this suggestion. We agree and have added a paragraph here in the methods to address this point, as follows:

*Though it is common in the literature to sample to 1 m deep in blue C sediments, the sampling protocol used for collecting these data (Ewers Lewis et al., 2018) was designed to maximize spatial coverage of shallow sediment C samples rather than sample entire sediment profiles (which may extend well beyond 1 meter deep). Greater spatial coverage allowed us to test the relationships between a variety of potential drivers and surface sediment C stocks on both fine and broad scales.*

**L154-155: would be good to report on the uncertainty associated with the use of spectroscopic techniques to estimate the 'C contents' of the samples. Any idea about the magnitude of this uncertainty? How were C stocks calculated? Were depth profiles of bulk density collected as well? Please briefly explain this, as this is important for the interpretation of the uncertainty on your results.**

Thank you, we appreciate this reminder to include a brief explanation here (in addition to referencing the original C stock data paper) to aid in the interpretation of our modelling results. We have added the following text to ensure transparency of this information in the present manuscript:

*Sediment C stocks to 30 cm deep (referred to throughout the paper as "shallow sediment C stocks") were estimated for 287 sediment cores from 96 blue C ecosystems across Victoria in southeast Australia*

*(Ewers Lewis et al., 2018; Figure 1). Full details of sample collection, laboratory analyses, and calculations of C stocks can be found in Ewers Lewis et al. (2018). Briefly, three replicate sediment cores (5-cm inner diameter) were taken in each ecosystem (n=125 in tidal marsh, n=60 in mangroves, and n=102 in seagrasses). Once back in the laboratory, samples were taken from three depths (0-2, 14-16, 28-30 cm) within each core. Samples were dried at 60°C until a consistent weight was achieved, then ground. Dry bulk density (DBD) was calculated as the dry weight divided by the original volume for all samples.*

*Based on the protocols by Baldock et al. (2013), a combination of diffuse reflectance Fourier transform mid-infrared (MIR) spectroscopy and elemental analysis via oxidative combustion using a LECO Trumac CN analyzer was used to determine organic C contents of all samples. Previous studies have demonstrated the accuracy of using MIR to estimate organic C stocks of sediments (Baldock et al., 2013; Van De Broek and Govers, 2019; Ewers Lewis et al., 2018). MIR spectra were acquired for all samples, then a subset of 200 representative samples was selected based on a principle components analysis (PCA) of the MIR results utilizing the Kennard-Stone algorithm. Gravimetric contents of organic carbon were measured directly in the laboratory for the 200-sample subset (Baldock et al. 2013). A partial least squares regression (PSLR) was created using a Random Cross Validation Approach (Unscrambler 10.3, CAMO Software AS, Oslo, Norway) and used to build algorithms to predict square root transformed total carbon, total organic carbon, total nitrogen, and inorganic carbon for the entire dataset. The PSLR model was evaluated based on parameters from the chemometric analysis of soil properties (Bellon-Maurel et al., 2010; Bellon-Maurel and McBratney, 2011), and the relationship between measured and predicted values was assessed based on slope, offset, correlation coefficient (r), R-squared, the root mean square error (RMSE), bias, and the standard error (SE) of calibration (SEC) and validation (SEP; see Ewers Lewis et al., 2018 for full details). R-squared values for all square root transformed variables were ≥0.94.*

*Sediment C stocks were calculated based on Howard et al., 2014. Organic C density (mg C cm$^{-3}$) was calculated by multiplying organic C content (mg C g$^{-1}$) by DBD (g cm$^{-3}$). Linear splines were applied to each core to estimate C density for each 2 cm increment within the 30 cm core, then C densities for each interval (measured and extrapolated) were summed and converted to Mg C ha$^{-1}$ to estimate total stock down to 30 cm deep for each core location…*

**L237: I would refer to table S4 here; this will help the reader to understand how the models were constructed**
Thank you, we have added a reference to table S4 here as suggested.

**L244: it's not clear from the text how the 'averaged models' were obtained and what these exactly are, please explain this in more detail**
Thank you for this suggestion. We want to make sure the generation of the averaged models is very clear, so we have added text to the following portion of the methods section:
*The dredge products of each global model (i.e. models created from "dredging") were ranked using AICc and the best models (delta AICc <2) were used to produce averaged models (named based on the global model they were generated from, e.g. global model 7 -> dredged and averaged -> averaged model 7). Averaged models were produced using the model.avg function ('MuMIn' package v. 1.42.1; Barton, 2018). The parameter estimates for each averaged model represent the average of that parameter's values from the models in which the variable appeared (from within the subset AICc<2).*

To help clarify the rationale for the modelling approach we used, which is better for generating robust predictions when complex predictors are involved but cannot utilize standard methods for generating standard errors, we have added the following text to the beginning of section 2.3:

*To identify drivers of shallow sediment C stock variability and create the best predictive model of shallow sediment C stocks to 30 cm deep we utilized a multi-step process based on an information theoretic approach and multimodel inference (Figure 3). Traditional approaches have relied on identification of the "best" data-based model; however, information-theoretic approaches allow for more reliable predictions through utilization of multiple models, especially in cases where lower ranked models may be essentially as good as the best (Burnham and Anderson, 2002; Symonds and Moussalli, 2011). Further, information theoretic model selection has been demonstrated to provide significant advantages for explaining phenomena with more complex drivers (Richards et al., 2011). Here, we first looked broadly at our variables of interest by narrowing down to the best models containing all possible variables ("global" models, as explained below) using AICc (Akaike information criteria, corrected for small sample size) to explain the variability observed in the training dataset (70% of total C stock data; Symonds and Moussalli, 2011). From there, we identified which variables within the best global models best explained the observed variability in C stock data in order to remove unnecessary variables from the model equation (through the process of "dredging" and selecting the best subset, explained in detail below). The validity of removing unnecessary variables from the model is supported by the concept of parsimony, which suggests models more complicated than the best model provide little benefit and should be eliminated (Burnham and Anderson, 2002; Richards, 2008). The best subset of models generated from the global models ("dredge products") were selected based on delta AICc<2, which are viewed as essentially interchangeable with the best model (Symonds and Moussalli, 2011). Each subset of best models was used to generate an averaged model, which was tested by generating predictions of C stocks for a reserved (30%) subset of the dataset. The best performing model was used to generate a predictive map of C stocks to 30 cm deep for mapped blue C ecosystems in Victoria.*

**L298: it is not clear what you mean with 'intercept'**
Thank you for this comment. We have clarified the definition of 'intercept' and added some text to clarify the meaning of other model outputs:

*Parameter estimates from averaged models suggests dominant species/EVC was the most important predictor of shallow sediment C stocks, and was the only variable for which the 95% confidence interval of the estimates did not cross zero (Tables 2 and S7), suggesting a true effect of the variable on observed C stock variability (an estimate that included zero means there is potentially no impact of the variable on C stocks). Specifically, seagrasses P. australis, R. megacarpa, Z. muelleri, and Z. nigricaulis had shallow sediment C stocks significantly different than those of coastal tussock saltmarsh (assigned as the intercept in the model, or baseline dominant species/EVC for which to compare the effect of other dominant species/EVCs on C stocks), while all other tidal marsh EVCs, mangroves, and seagrass L. marina did not. This was confirmed by the ANOVA and Tukey's pairwise comparisons…*

**L333: Would be good to provide a measure of uncertainty on the total calculated C stock, similar to the standard deviations you report on the calculated numbers further down in this paragraph.**
Thank you for this suggestion. The standard deviation estimates you are referring to in the text were calculated by taking the average of the predicted C values for all individual cells overlapping with each ecosystem's areal extent, then taking the standard deviation. The model predictions are spatially explicit; i.e. a predicted C value is generated for each individual raster cell based on the unique characteristics (i.e.

combination of spatial data) of that cell that were included in the averaged model. Therefore, it was not possible to predict a single standard deviation or standard error for the total C stock for the entire state of Victoria in the same way because it was a sum.

Instead, the uncertainty of our predicted C stock estimates is represented in our validation step (e.g. Figure S3). We reserved 30% of our C stock dataset to test the accuracy of the predictions generated by each averaged model (as described in sections 3.2 of the results and displayed in Figure S3). These results showed that our best model explained ~half the observed variability in C stocks (Adj R-sq=0.4881; averaged model 2). We have clarified the errors associated with our model predictions by providing further details on the outputs of our validation step, which utilized the reserved dataset (30% of our original carbon dataset) for assessing the prediction power of each averaged model. Using linear regression, we compared predicted values from each averaged model to actual measured C stock values for reserved dataset. The complete outputs for this analysis have been added to the results section 3.2 ("Model validation"), as follows:

*Linear regressions of predicted versus actual measured shallow sediment C values produced the following outputs for each averaged model: averaged model 11, residual standard error (RSE)=38.36 on 84 degrees of freedom (df), adjusted R-squared (R-sq(adj))=0.4868, F-statistic(F-stat)=81.63 on 1 and 84 df, p-value=5.044e-14; averaged model 5, RSE=38.51, R-sq(adj)=0.4829, F-stat=80.39 on 1 and 84 df, p-value=6.953e-14; averaged model 2, RSE=38.32, R-sq(adj)=0.4881, F-stat=82.06 on 1 and 84 df, p-value=4.517e-14; averaged model 10, RSE=39.67, R-sq(adj)=0.4514, F-stat=70.93 on 1 and 84 df, p-value=8.645e-13; averaged model 4, RSE=39.84, R-sq(adj)=0.4465, F-stat=69.58 on 1 and 84 df, p-value=1.254e-12; averaged model 1; RSE=39.48, R-sq(adj)=0.4566, F-stat=72.43 on 1 and 84 df, p-value=5.73e-13; averaged model 7, RSE=39.29, R-sq(adj)=0.4618, F-stat=73.94 on 1 and 84 df, p-value=3.81e-13.*

We have also been more explicit about the error associated with the averaged model used to generate our state-wide shallow sediment C stock predictions by adding text to the results section on modelled shallow sediment blue C stocks (section 3.3) as follows:

*We estimated a total of over 2.31 million Mg C stored in the top 30 cm of sediments in the ~68,700 ha of blue C ecosystems across Victoria (Table 4; Figure 5). This estimate is based on predictions from our best averaged model that utilized ecosystem type as the ecological variable (averaged model 7), which explained 46.18% of observed variability in C stock data and had an RSE of 39.29.*

Due to the complexity of our multimodel approach, we did not include standard errors in predicted values for the entire region of Victoria due to the combination of using an averaged model and having random effects (there are no compatible R packages, to our knowledge).

**L336: Please briefly explain how the standard deviations were calculated. What do they exactly represent? Only the spatial variation within these ecosystems, or also uncertainties related to the model procedures used?**
The standard deviations here refer to those related to the mean of predicted C stock values for every raster cell of each ecosystem's mapped areal extent. We have updated the text, as follows, to clarify this:

*Mean predicted C stocks (±SD) to 30 cm deep for each ecosystem type were 57.96 (±2.90) Mg C ha$^{-1}$ for tidal marsh, 50.64 (±1.35) Mg C ha$^{-1}$ to mangroves, and 23.48 (±0.57) Mg C ha$^{-1}$ for seagrass based on predicted C stock values in all raster cells of each ecosystem's mapped areal extent in Victoria. These C*

*stock values ranged from 23.33 – 291.18, 23.34 – 77.81, and 23.33 – 73.42 Mg C ha$^{-1}$ for tidal marsh, mangroves, and seagrass, respectively.*

**L411: I would suggest changing this title to 'Modelled topsoil blue C stocks'**
Thank you, we have changed it to "Modelled **shallow sediment** blue C stocks" here and in the results to be consistent with the wording in the text body and to be transparent about the depth of sediments considered in the study.

**Tables and figures**

**Table 2 and 3: it would be good to refer to where the description of the global models can be found (Table S4) in the caption (after '(global model 11, 5, 2 and 8)')**
Thank you for this suggestion, we have added a reference to the supplementary Table S4 in each of the two table captions (Tables 2 and 3) as you have suggested.

**Table 4: I would change the caption to: '[. . .] and calculated C stocks [. . .]'**
We have changed this as suggested, thank you.

**Table 5: I would change the caption to: 'Calculated blue C stocks [. . .]'**
We have changed this as suggested, thank you.

**Figure 4: Please explain in the caption what the error bars represent (standard deviation?) and how they should be interpreted.**
Thank you for pointing out that this information was missing from the caption. We have updated the Figure 4 caption to more clearly describe how the figure should be interpreted, including the meaning of the error bars, as follows:

*Figure 4. Measured C stocks (Mg C ha$^{-1}$; average ± standard error) in the top 30 cm of sediment by dominant species/ecological vegetation class (EVC). Bars are color-coded by ecosystem type: red = tidal marsh, green = mangrove, blue = seagrass. C stocks differed significantly by dominant species/EVC, with higher C stocks in coastal tussock saltmarsh, wet saltmarsh herbland, wet saltmarsh shrubland, mangroves A. marina, and seagrass L. marina (group a) compared to seagrasses P. australis, Z. nigricaulis, and Z. muelleri (group b; ANOVA and Tukey pairwise comparison, $F_{8,284}$ = 34.80, p < 0.001, R-sq(adj) = 48.77 %). Error bars represent one standard error of the mean.*

**Technical corrections**

**L40: remove 'our'**
We have corrected this, thank you.

**L250: variable => variables**
We have corrected this, thank you.